# Measurement of exclusive $J/\psi$ and $\psi(2S)$ production at $\sqrt{s} = 13$ TeV

**LHCb collaboration**$^\star$

## Abstract

Measurements are presented of the cross-section for the central exclusive production of $J/\psi \rightarrow \mu^+\mu^-$ and $\psi(2S) \rightarrow \mu^+\mu^-$ processes in proton-proton collisions at $\sqrt{s} = 13$ TeV with 2016–2018 data. They are performed by requiring both muons to be in the LHCb acceptance (with pseudorapidity $2 < \eta_{\mu^\pm} < 4.5$) and mesons in the rapidity range $2.0 < y < 4.5$. The integrated cross-section results are

$$\sigma_{J/\psi \rightarrow \mu^+\mu^-}(2.0 < y_{J/\psi} < 4.5, 2.0 < \eta_{\mu^\pm} < 4.5) = 400 \pm 2 \pm 5 \pm 12 \,\text{pb},$$
$$\sigma_{\psi(2S) \rightarrow \mu^+\mu^-}(2.0 < y_{\psi(2S)} < 4.5, 2.0 < \eta_{\mu^\pm} < 4.5) = 9.40 \pm 0.15 \pm 0.13 \pm 0.27 \,\text{pb},$$

where the uncertainties are statistical, systematic and due to the luminosity determination. In addition, a measurement of the ratio of $\psi(2S)$ and $J/\psi$ cross-sections, at an average photon-proton centre-of-mass energy of 1 TeV, is performed, giving

$$\frac{\sigma_{\psi(2S)}}{\sigma_{J/\psi}} = 0.1763 \pm 0.0029 \pm 0.0008 \pm 0.0039,$$

where the first uncertainty is statistical, the second systematic and the third due to the knowledge of the involved branching fractions. For the first time, the dependence of the $J/\psi$ and $\psi(2S)$ cross-sections on the total transverse momentum transfer is determined in $pp$ collisions and is found consistent with the behaviour observed in electron-proton collisions.

## Contents

$^\star$ Authors are listed at the end of this paper.

## 1 Introduction

Deep inelastic scattering of leptons off protons provided the first proof that hadrons are not elementary but rather composed of quarks [1, 2]. It is an essential tool to determine parton distribution functions (PDFs) inside protons, which are required to make cross-section predictions at hadron colliders. However, charged leptons interact electromagnetically and only probe the density of the quarks, which are charged. The densities of the neutral gluons must be inferred, which can be done by studying how the quark PDFs evolve with the scale set by the mass of the exchanged virtual photon. These PDFs are determined in fits [3–5] to multiple measurements, including notably $e^{\pm}p$ scattering [6, 7], and forward production of vector bosons [8–11] and heavy-quarks [12–15] in $pp$ collisions. Due to a lack of data at low $x$, the fraction of hadron momentum carried by the parton, the uncertainties attributed to the gluon PDFs are large at low $x$ and are even compatible with an unphysical decrease of the gluon density with $x$ [16]. Other methods are thus required to access the gluonic PDF.

Central exclusive vector-meson production (CEP) in $pp$ collisions is the quasi-elastic production of a single meson, leaving the protons intact. Exclusive charmonium production results from the conversion of a virtual photon close to its mass shell into a $c\bar{c}$ pair, which hadronises into a $J/\psi$ or $\psi(2S)$ meson. These processes probe the gluonic PDF at the scale of the charm quark mass. The exclusivity of the process requires that, at leading order, two gluons are exchanged with the target hadron. Thus the cross-section approximately scales as gluon density squared [17–20]. The process and the main backgrounds are depicted in Fig. 1.

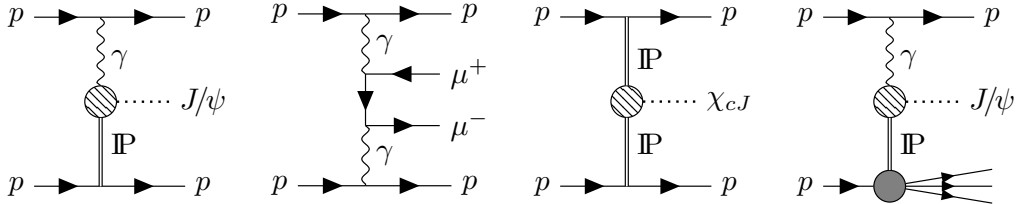

Figure 1: Feynman diagrams for signal and background processes. From left to right: signal CEP $J/\psi$ photoproduction, where $\mathbb{P}$ stands for a colourless superposition of gluons, sometimes referred to as a pomeron; continuum dimuon production; exclusive $\chi_c$ ($J = 0, 1, 2$) production via double pomeron exchange; inelastic $pp$ collision where a proton dissociates.

Exclusive scattering processes also give access to the total transverse momentum transfer $\Delta t$, the square of the difference between the momenta of the incoming and outgoing proton, which is a Fourier conjugate to the impact parameter between two colliding hadrons. As such, the $\Delta t$ spectra are sensitive to the spatial distribution of colour charge [21]. Several predictions (see *e.g.* Ref. [18]) calculate the exclusive production cross-section at $\Delta t \sim 0$. The cross-section falls exponentially versus $\Delta t$ with a slope determined experimentally, which can be used to infer the total exclusive cross-section. In the present paper, this slope is determined in ten intervals of rapidity for the $J/\psi$ and $\psi(2S)$ mesons. As the outgoing protons are not detected at LHCb, $\Delta t$ is not directly accessible and the transverse momentum squared, $p_T^2$, of the charmonium state is used as a proxy.

The photoproduction cross-section of a charmonium state is sensitive to the radial wave function of the charmonium state in a region where the $\psi(2S)$ wave function has a radial node but the $J/\psi$ wave function does not. As a result, the $\psi(2S)$ photoproduction cross-section is expected to be suppressed with respect to that of $J/\psi$ mesons [22–31]. With many theoretical uncertainties cancelling, predictions for the ratio of $\psi(2S)$ and $J/\psi$ cross-sections can be determined more precisely than the individual cross-sections.

Exclusive $J/\psi$ and $\psi(2S)$ production in $pp$ collisions at the LHC have previously been measured at centre-of-mass energies of $\sqrt{s} = 7$ TeV [32,33] and 13 TeV [34]. Exclusive double-charmonium [35] and $\Upsilon$ [36] production have been measured at 7 and 8 TeV, and that of $J/\psi\phi$ at 13 TeV [37]. Charmonia production has also been studied in ultra-peripheral $p$Pb [38] and PbPb [39–42] collisions.

The previous LHCb measurements have been used to update PDF fits [19,43], and thus improve predictions of $J/\psi$ and $\Upsilon$ CEP cross-sections [18,44,45]; make predictions [46–52] for ultra-peripheral photoproduction processes at RHIC [53,54] and the LHC [41,42,55,56]; determine the meson-proton scattering length [57] and extract the proton mass radius from the $J/\psi$ and $\psi(2S)$ cross-sections [58]. Based on these cross-sections, Ref. [59] claims that LHCb data show evidence of gluon saturation, *i.e.* the slowing down of the growth of gluon densities as $x$ decreases due to gluon emission and recombination balancing each other, while the authors of Ref. [45] disagree. Such effects would usually be expected in heavy-ion collisions.

This paper presents a measurement of exclusive $J/\psi$ and $\psi(2S)$ production in proton-proton collisions at $\sqrt{s} = 13$ TeV in the forward direction, in ten intervals of rapidity between 2.0 and 4.5. The data used were collected with the LHCb detector at the LHC between 2016 and 2018, corresponding to an integrated luminosity of $4.4\,\text{fb}^{-1}$, which is twenty times larger than that used in Ref. [34]. This larger sample permits a better control of background shapes, implemented in a two-dimensional fit in dimuon mass and transverse-momentum squared. For the first time, a measurement of the $\psi(2S)$ cross-section in the same rapidity intervals as for the $J/\psi$ cross-section, and thus the determination of their ratio as a function of rapidity is presented.

## 2 Detector, simulation and data sample

The LHCb detector [60,61] is a single-arm forward spectrometer covering the pseudorapidity range $2 < \eta < 5$, designed for the study of particles containing $b$ or $c$ quarks. The detector includes a high-precision tracking system consisting of a silicon-strip vertex detector (VELO) surrounding the $pp$ interaction region [62], a large-area silicon-strip detector located upstream of a dipole magnet with a bending power of about $4\,\text{T m}$, and three stations of silicon-strip detectors and straw drift tubes [63] placed downstream of the magnet. The tracking system provides a measurement of the momentum, $p$, of charged particles with a relative uncertainty

that varies from 0.5% at low momentum to 1.0% at 200 GeV/$c$. Photons are identified by a calorimeter system consisting of scintillating-pad and preshower detectors (SPD), and electromagnetic and hadronic calorimeters. Muons are identified by a system composed of alternating layers of iron and multiwire proportional chambers [64].

The pseudorapidity coverage of the LHCb detector is extended by the HERSCHEL system, composed of forward shower counters consisting of five planes of scintillators with three planes at 114, 19.7 and 7.5 m upstream of the interaction point, and two downstream at 20 and 114 m. At each location, there are four quadrants of scintillators, whose information is recorded in every beam crossing by photomultiplier tubes, giving a total of 20 channels in HERSCHEL [65]. These are calibrated using data taken without beams circulating at the end of each LHC fill [66]. The pseudorapidity ranges covered by the VELO and HERSCHEL are different. For the VELO the region is $-3.5 < \eta < -1.5$ and $2 < \eta < 5$, and for HERSCHEL the region is $-10 < \eta < -5$ and $5 < \eta < 10$.

The online event selection is performed by a trigger [67,68] that consists of a hardware stage, based on information from the calorimeter and muon systems, followed by a software stage, which applies a full event reconstruction. The distinct signature of CEP events is their low multiplicity. Consequently, at the hardware stage, the trigger selects events containing at least one muon with $p_T > 192$ MeV/$c$ and fewer than 20 hits in the SPD detector. At the software stage events are selected if they contain two muons with $p_T > 400$ MeV/$c$, fewer than 10 tracks in the VELO, of which at most four are reconstructed in the backward direction [62]. A sample used for the determination of trigger, reconstruction and particle identification (PID) efficiencies is collected requiring a single muon with $p_T > 500$ MeV/$c$ and the same multiplicity requirements as for the default selection.

The data used were collected between July 2016 and October 2018. The early 2016 data are not used as relevant trigger selections were not yet included. Data from the last month of data taking in 2018 is also discarded as it was affected by a noisy SPD readout board, which biases the number of SPD hits in low-multiplicity events.

Offline, events are required to contain only the two muon candidates, which should be of good quality [69], and identified as such [70], which implies that their momentum exceeds 3 GeV/$c$, the threshold to cross the calorimeter and reach the muon system. The event should contain no additional tracks in the VELO, and no photons other than those that are consistent with being radiated from the passage of muons through the detector material.

The muons from CEP signal $J/\psi$ decays are well outside of the HERSCHEL acceptance; these counters are used to veto charged particles from the proton dissociating. The CEP cross-section measurements are performed with events that contain no such additional particles, *i.e.* HERSCHEL signals consistent with noise. The remaining events are retained for background studies. The HERSCHEL response is described using a discriminating $\chi^2$-like variable that quantifies the activity above noise taking into account correlations between the counters [65]. The selection requirement is optimised using low-mass low-$p_T^2$ dimuon pairs, which are dominated by two-photon fusion.

Simulation is required to model the effects of the detector acceptance and the imposed selection requirements, and to study specific backgrounds. In the simulation, the charmonium candidate is generated and decayed using SuperChic2 [71], with the exception of $\psi(2S) \rightarrow J/\psi X$ processes (where $X$ is any combination of particles, mostly $\pi\pi$), for which the decay is handled by EVTGEN [72]. Final-state radiation is generated using PHOTOS [73]. The interaction of the generated particles with the detector, and its response, are implemented using the GEANT4 toolkit [74,75] as described in Ref. [76]. The ROOT [77] and LHCb [78–80] software frameworks are used for the initial data preparation, while the analysis is written in the PYTHON language with standard scientific packages [81–86].

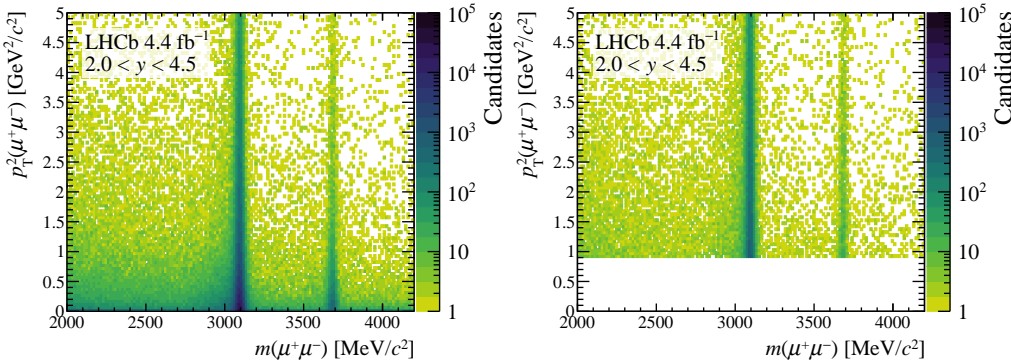

Figure 2: Two-dimensional mass-$p_T^2$ distributions for the (left) signal and (right) control samples.

The total integrated luminosity of the used data sample is determined using empty-event counters calibrated by van der Meer scans and beam profile measurements [87] and is found to be $\mathcal{L}_{\text{int}} = 4.41 \pm 0.13\,\text{fb}^{-1}$. Due to the multiplicity requirements imposed in the trigger and the offline selection, only events with a single $pp$ interaction are selected. The useful integrated luminosity is thus reduced by the fraction of events with a single interaction containing at least two VELO tracks. The number of such visible $pp$ interactions per beam crossing, $n$, is assumed to follow a Poisson distribution, $P(n) = \mu^n e^{-\mu}/n!$, with mean $\mu$. The fraction of useful integrated luminosity, $\mathcal{L}_{\text{int}}^{\text{eff}}$, corresponding to events with $n = 1$, is given by

$$f_{\mathcal{L}} = \frac{\mathcal{L}_{\text{int}}^{\text{eff}}}{\mathcal{L}_{\text{int}}} = \frac{P(n=1)}{\sum_{n=0}^{\infty} nP(n)} = \frac{\mu e^{-\mu}}{\sum_{n=0}^{\infty} n\frac{\mu^n e^{-\mu}}{n!}} = e^{-\mu}. \tag{1}$$

The value of $\mu$ depends on running conditions and it is determined in periods of up to one hour of stable running conditions [87]. In most running periods $\mu$ is close to 1.1, with variations of less than 10%, corresponding to an average $f_{\mathcal{L}} \simeq 0.33$. The corresponding useful integrated luminosity is $\mathcal{L}_{\text{int}}^{\text{eff}} = 1522 \pm 44\,\text{pb}^{-1}$, where the uncertainty is dominated by that on $\mathcal{L}_{\text{int}}$.

## 3 Two-dimensional signal fits

The primary challenge in this analysis is separation of the elastic CEP and inelastic proton-dissociation (PD) components, shown in Fig. 1. The latter consists of events where the proton dissociates, producing charged particles in the very forward acceptance. These are vetoed by the HERSCHEL requirement, which however is not perfect and thus leaves some PD backgrounds in the selected signal sample. The different $p_T^2$ distributions of PD and CEP charmonia are therefore also exploited. The properties of the PD component are determined from a control sample that is free from any CEP signal contribution. This sample is obtained by inverting the HERSCHEL veto, and requiring $0.9 < p_T^2 < 5.0\,\text{GeV}^2/c^2$, where the CEP contribution, which populates the low-$p_T^2$ region, is negligible.

Two other backgrounds are accounted for: QED continuum dimuon production and $J/\psi$ feed-down from higher-mass charmonia, namely $\psi(2S)$, and $\chi_{cJ}(1P)$ ($J = 0, 1, 2$), referred to as $\chi_c$ below unless otherwise specified. Other feed-down contributions, such as those from $\Upsilon$ resonances or $b$ hadrons, are negligible because of the VELO-tracks veto.

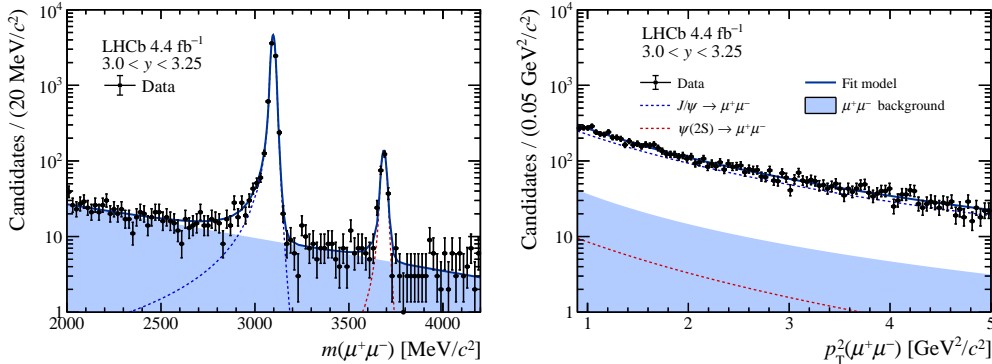

Figure 3: Distributions of (left) mass and (right) $p_T^2$ of data in the control sample for rapidity interval $3.0 < y < 3.25$. The fit described in the text is superimposed.

As there is a correlation between the dimuon mass and $p_T^2$ distributions for non-peaking backgrounds, the data are fit by a two-dimensional model in mass and $p_T^2$ in each of ten rapidity intervals. The considered regions are $2000 < m_{\mu^+\mu^-} < 4200 \,\mathrm{MeV}/c^2$ and $p_T^2 < 5 \,\mathrm{GeV}^2/c^2$. The relevant distributions are shown in Fig. 2. Overall, there are 566 095 events in the signal sample and 56 654 in the control sample.

The signal yields are determined in each rapidity interval by a two-dimensional unbinned extended maximum-likelihood fit [83, 84] in mass and $p_T^2$. The fit model comprises the signal $J/\psi$ and $\psi(2S)$ components; the continuum QED background; the $\psi(2S)$ and $\chi_c$ feed-downs; and the inelastic PD background. Prior to carrying out the fit in each rapidity interval, fits to the whole signal and control data samples (referred to as the full sample) are performed and their results are used to constrain nuisance parameters that cannot be determined accurately in low-yield rapidity regions, as described below.

The CEP and PD $J/\psi$ and $\psi(2S)$ mass peaks are each modelled with a Gaussian function, modified to have power-law tails on both sides [88]. The difference in the means of the two Gaussian components is fixed according to the known mass difference of the two resonances [89]. Their widths are constrained to scale linearly with the energy release in the respective decays [90]. The tail parameters are shared between the two peaks and Gaussian-constrained to the values determined in the fits to the full sample.

The CEP $p_T^2$ shape is independent of the mass and is described by an exponential function, as expected by Regge theory [91] and measured in previous experiments, notably at HERA [92]. The slopes of the $J/\psi$ and $\psi(2S)$ exponentials are left free to float in each rapidity interval.

The $p_T^2$ distribution of the PD $J/\psi$ and $\psi(2S)$ mesons is modelled with a power-law function proportional to $(1 + (b_{pd}/n_{pd})p_T^2)^{-n_{pd}}$, as measured by the H1 experiment [93]. This function follows approximately an exponential of slope $-b_{pd}$, modified by the empirical parameter $n_{pd}$. Alternative models are discussed in Sec. 5. In addition, the PD contribution contains a nonresonant component which is modelled by an exponential shape in mass and the above-mentioned power-law model for $p_T^2$. The parameters of the three power-laws are different for the $J/\psi$, $\psi(2S)$, and nonresonant dimuon components.

The parameters of the PD components are first determined by a fit to the control sample in each rapidity interval; an example fit is shown in Fig. 3. All parameters are free to vary in these fits except for the signal tail parameters, as explained above.

The PD models are then used as input in the fits to the signal sample. The $p_T^2$ shapes of the PD components are fixed to the values obtained on the corresponding control sample, while the PD $J/\psi$ and $\psi(2S)$ mass shapes are forced to be identical to those of the CEP signals. The

relative fractions of the two charmonia and the dimuon background are constrained from the fit to the control sample.

Exclusive continuum, or nonresonant dimuon production, is a QED process that takes place via the fusion of two photons. The dimuon pair produced in this form has low dimuon mass and a $p_\mathrm{T}^2$ shape sharply peaked towards zero. The mass and $p_\mathrm{T}^2$ distributions are correlated and therefore a two-dimensional histogram, which is obtained from simulation and validated with low-$p_\mathrm{T}^2$ data, is used in the fit.

The $J/\psi$ yield is affected by feed-down from higher-mass charmonium states, which is accounted for in the fit. The feed-down from $\psi(2S) \to J/\psi X$ decays is partially suppressed by the VELO and SPD multiplicity requirements. The yield of the remaining feed-down is determined from simulation of inclusive $\psi(2S) \to J/\psi X$ processes, normalised by the $\psi(2S)$ yield measured in each rapidity interval. Bin migration is taken into account via a migration matrix determined from simulation. An iterative procedure is applied to first determine the rapidity-dependent $\psi(2S)$ yield and then its contribution to the $J/\psi$ yield. In practice, two steps are sufficient for the convergence of the procedure.

The normalisation of the feed-down from $\chi_c \to J/\psi \gamma$ decays is determined by reconstructing $J/\psi \gamma$ candidates in data. The same $J/\psi$ selection as for CEP and PD candidates is used, except that the veto on additional photons is removed. Instead, photons with transverse energy in excess of 75 MeV are combined with $J/\psi$ candidates to form $\chi_c$ candidates. In each $y$ interval, where $y$ is the rapidity of the $J/\psi$ meson, and separately for the signal and control samples, the $\chi_{cJ}$ ($J = 0, 1, 2$) yields are determined from a fit to the resulting mass distribution. Fits to the $\chi_c$ samples are shown in Fig. 4. The three $\chi_{cJ}$ ($J = 0, 1, 2$) mass peaks are each modelled with a Crystal Ball function [88] with the tail parameters fixed from simulation. The peak of the Gaussian is free in the fit to account for imperfect photon energy calibration, but the shift with respect to the known masses of the $\chi_c$ mesons [89] is constrained to be the same for all three states. The shift varies between 6 and 10 MeV/$c^2$ (with typical statistical uncertainties between 0.5 and 1 MeV/$c^2$) depending on the rapidity interval. The background is a mixture of partially reconstructed $\psi(2S)$ decays, such as $\psi(2S) \to J/\psi \pi^0 \pi^0$, and random combinations of $J/\psi$ mesons and calorimeter clusters. The same empirical function as in Ref. [94] is used as a model for the sum of these contributions.

The contribution from $\chi_{c0}$ mesons is small, while those of $\chi_{c1}$ and $\chi_{c2}$ mesons dominate. The empirical background model does not describe perfectly the mass distribution in the region below 3400 MeV/$c^2$, which has a negligible effect on the total $\chi_c$ yield. Due to the limited photon energy resolution, the mass fit has little sensitivity to the relative size of the $\chi_{c1}$ and $\chi_{c2}$ contributions. This ambiguity however does not affect the determination of the total $J/\psi$-from-

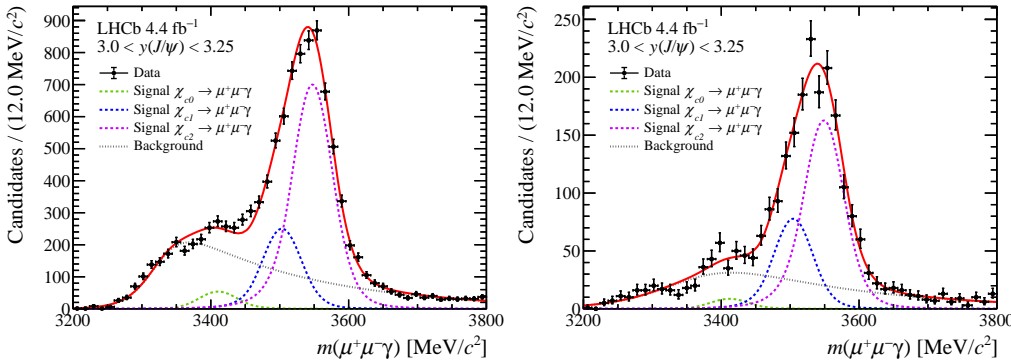

Figure 4: Fit to the $J/\psi \gamma$ mass distribution with the $J/\psi$ meson in $3.0 < y < 3.25$ for (left) signal and (right) control samples.

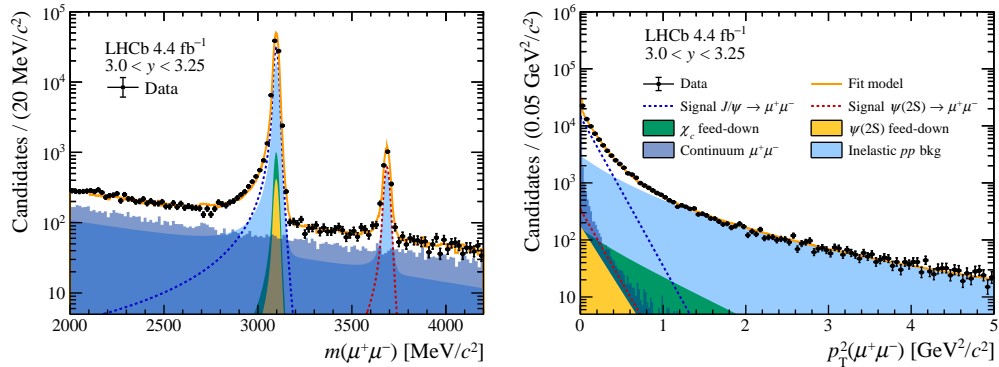

Figure 5: Distributions of (left) mass and (right) $p_T^2$ of data in the signal sample for the rapidity interval $3.0 < y < 3.25$. The fit described in the text is superimposed.

$\chi_{c(1,2)}$ yield since *(i)* the branching fractions of the $\chi_{c(1,2)} \to J/\psi\gamma$ decays drop out in the ratio of $J/\psi$ to $\chi_{c(1,2)}$ yields; *(ii)* the relative efficiencies for reconstructing $J/\psi\gamma$ and $J/\psi \to \mu^+\mu^-$ are equal for $\chi_{c1}$ and $\chi_{c2}$, as determined from simulation; and *(iii)* the $p_T^2$ distributions of $J/\psi$ from $\chi_{c1}$ and $\chi_{c2}$ are found to be equal in simulation and in data, which is checked by investigating the $p_T^2$ shape of candidates in the left and right halves of the $\chi_{c(1,2)}$ mass peak. The $J/\psi$-from-$\chi_{c(1,2)}$ yield is therefore proportional to the $\chi_{c(1,2)}$ yield in each rapidity interval. This feed-down contribution is determined in the signal and control samples, and the PD contribution is subtracted from that in CEP events to determine the overall CEP $\chi_c$ feed-down normalisation.

The $J/\psi$-from-$\psi(2S)$ and $J/\psi$-from-$\chi_c$ components are modelled in the CEP fit using the same mass model as for the $J/\psi$ signal. The $p_T^2$ shapes are modelled with a single (double) exponential distribution for the $\psi(2S)$ ($\chi_c$) feed-down, which is determined from simulation that is validated by data.

The mass and $p_T^2$ projections of the fit in the interval $3.0 < y < 3.25$ are shown in Fig. 5. All intervals are shown in Fig. 11 in Appendix C. The parameters of interest are the CEP $J/\psi$ and $\psi(2S)$ yields, and the slopes of their $p_T^2$ shapes. In total, $299\,100 \pm 2100$ $J/\psi$ and $7420 \pm 130$ $\psi(2S)$ elastically produced mesons are found in the fit to the full rapidity range.

## 4 Efficiencies

The signal yields are corrected for detection efficiencies using simulation samples calibrated with data [62, 69, 70], except for the HeRSCheL-related efficiencies, which are estimated in data.

A tag-and-probe method, aimed at measuring single-muon efficiencies, is applied to account for the differences between simulation and data. The simulation sample is then weighted with the appropriate correction factors [66]. In this method, a tag muon from the $J/\psi$ candidate is required to pass all selection criteria, while the other muon is used to measure the efficiency under investigation. The same procedure is applied to calibration samples and simulation, and the latter is weighted by the ratio of those efficiencies. The tracking, PID and hardware muon trigger efficiencies are calibrated in this manner. As most efficiencies depend on muon kinematics, they are determined in regions of muon pseudorapidity and transverse momentum, and separately for each year of data taking. Depending on the considered $p_T, \eta$ region, correction factors range between 0.9 and 1.1 for tracking, 0.8 and 1.2 for PID, and 0.7 and 1.1 for muon trigger efficiencies, with uncertainties between 1% and 3%.

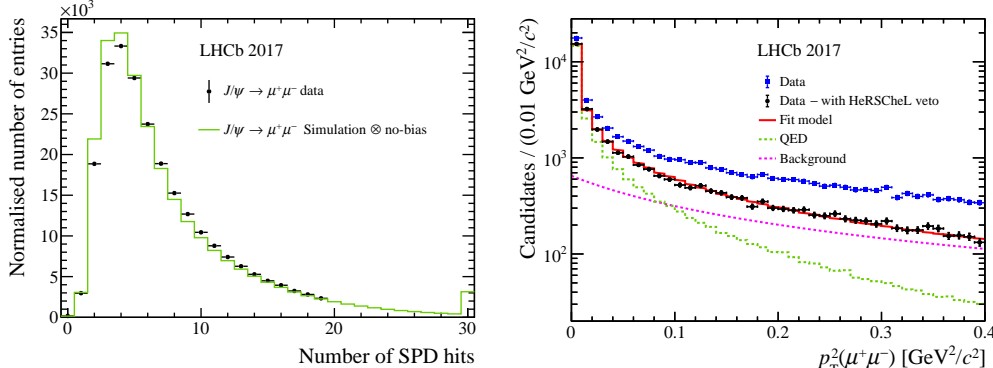

Figure 6: (Left) SPD multiplicity distributions in 2017 $J/\psi \to \mu^+\mu^-$ data and modelling with simulation corrected using no-bias data from events with no beam crossing. The last bin contains the overflow events with 30 SPD hits or more. (Right) $p_T^2$ distributions in 2017 data without and with the HERSCHEL requirement applied. The fit is only shown for the latter distribution.

The hits in the SPD detector are due to charged particles reaching the detector, including those produced by the preshower detector, and to spill-over from the previous $pp$ interaction. In the case of CEP events, which have only two muon tracks, the latter component dominates; however, it is not well modelled in simulation. The SPD hit distribution due to spill-over is obtained from data events that were collected by random triggers in unfilled bunch crossings that followed bunch crossings with a collision. This sample is referred to as no-bias data in the following. The obtained distribution is convolved with the SPD multiplicity in $J/\psi \to \mu^+\mu^-$ simulation and matches sufficiently well the distribution observed in CEP data, especially the tail up to the cut value of 20 SPD hits, as shown in Fig. 6. The effect of the remaining mismodelling is addressed in Sec 5. The fraction of events above this value defines the SPD trigger inefficiency, which is found to be independent of the dimuon kinematics.

The HERSCHEL detector is not included in the simulation. Its efficiency is determined using dimuon QED events, with and without the HERSCHEL vetoes applied. The $p_T^2$ distributions are shown in Fig. 6, emphasising the fact that the HERSCHEL requirement has little effect at vanishing $p_T^2$, where QED backgrounds dominate. The efficiency is determined from the ratio of the QED components determined by the fits to the distributions with and without the HERSCHEL veto applied. It is found to be between 85% and 90% depending on data-taking period.

Efficiencies for the requirement on the absence of additional VELO tracks or photons are taken from simulation and cross-checked in data in the same way as for the HERSCHEL veto efficiency. They are close to unity. The software trigger is fully efficient with respect to the offline selection. The total efficiency varies between 40% and 55%, with the lowest values being at the edges of the rapidity acceptance.

## 5  Systematic uncertainties

Systematic uncertainties arise due to the size of the simulation and calibration samples, and from the luminosity determination; they are mentioned in the sections above. Systematic uncertainties related to modelling choices are described below. All values are listed in Tables 1 and 2.

In the modelling of the number of SPD hits, $N_{\text{SPD}}$, the convolution of the no-bias data and the $J/\psi \to \mu^+\mu^-$ simulation is normalised to the distribution seen in data for $N_{\text{SPD}} < 20$. Alternatively, one could normalise it to get the best possible description of the tail, *i.e.* in the region $8 \le N_{\text{SPD}} < 20$. The resulting change in SPD veto efficiency is reported as a systematic uncertainty.

The efficiencies of the HERSCHEL, track, and photon vetoes, collectively called global event cuts (GEC) in Table 1, are taken from the yield of QED events passing the veto. The fit models the non-QED background with the power-law from Ref. [93], but in this reduced $p_{\text{T}}^2$ region the sum of two exponential functions also provides a good description. The difference in measured efficiencies is taken as the associated systematic uncertainty.

The signal $J/\psi$ and $\psi(2S)$ mass peaks are described by a modified Gaussian function. Other models, such as the sum of a Gaussian and a Crystal Ball function [88], are tried but yield poor-quality fits, without however significantly affecting the signal yields. No uncertainty is assigned. The peak value of the $\psi(2S)$ mass shape is constrained to be offset from the $J/\psi$ peak by the known mass difference of the two mesons. A systematic uncertainty is estimated by scaling the offset proportionally to the energy release in each decay.

The $p_{\text{T}}^2$ shapes are modelled by a single exponential, as expected from theory, and there is no evidence of the need for another model. The $p_{\text{T}}^2$ shapes of the $J/\psi$ from feed-down are modelled with exponential functions taken from simulation. Alternatively, nonparametric distributions obtained from simulation are used, which yields a small change in the feed-down contributions and thus the signal yields. For the $\chi_c$ feed-down, a determination of the contribution from each $\chi_c$ state would be needed if their $p_{\text{T}}^2$ distributions were different. The present data do not require this. A dedicated CEP $\chi_c$ study using converted photons would be needed to resolve the $\chi_{c1}$ and $\chi_{c2}$ states.

The mass distribution of the inelastic $pp$ background is described by the same shape as for the signal $J/\psi$ and $\psi(2S)$ plus an exponential to describe the nonresonant dimuon contribution. A systematic uncertainty is estimated by changing the single exponential to the sum of two exponential functions.

Similarly, the $p_{\text{T}}^2$ shape for each component of the inelastic $pp$ background is modelled with a power law, as measured by the H1 experiment. However, the fit to the H1 dataset is not perfect at very low $p_{\text{T}}^2$ [93]. Therefore other models are investigated. The sum of two exponentials is found to also provide a reasonable fit, though of slightly lower quality. The systematic uncertainty due to this modelling is determined with pseudoexperiments. In each rapidity interval the data are fit with the alternate mass-$p_{\text{T}}^2$ model and then 500 pseudodata samples are generated using the fit result as model. These pseudosamples are then fit with the default model. The resulting biases of the $J/\psi$ and $\psi(2S)$ yields are assigned as a systematic uncertainty.

Fit biases are tested in the same way. This time the default shape is used to generate 500 pseudodata samples which are fit with the same model. The variation of yields is compatible with the uncertainty returned from the fit, and the biases are negligible in comparison. No uncertainty is assigned.

Table 1 shows the values of all the systematic uncertainties previously discussed per rapidity interval for the $J/\psi \to \mu^+\mu^-$ and $\psi(2S) \to \mu^+\mu^-$ cross-section measurements. All uncertainties are assumed to be uncorrelated between rapidity intervals, except those for the SPD and HeRSCHeL multiplicities, and for the luminosity, which are fully correlated.

The slope of the signal $p_{\text{T}}^2$ shape is only affected by changes in the fit model, leading to the systematic uncertainties listed in Table 2.

Table 1: Systematic and statistical uncertainties per rapidity interval for the $J/\psi$ and $\psi(2S)$ cross-section in percent. The luminosity uncertainty is listed separately. Values below 0.005% are not shown.

| $y$ bin Source / state | 2.0–2.25 $J/\psi$ | $\psi(2S)$ | 2.25–2.5 $J/\psi$ | $\psi(2S)$ | 2.5–2.75 $J/\psi$ | $\psi(2S)$ | 2.75–3.0 $J/\psi$ | $\psi(2S)$ | 3.0–3.25 $J/\psi$ | $\psi(2S)$ |
|---|---|---|---|---|---|---|---|---|---|---|
| **Uncorrelated uncertainties** | | | | | | | | | | |
| Simulation sample size | 0.35 | 0.6 | 0.16 | 0.28 | 0.12 | 0.21 | 0.10 | 0.18 | 0.10 | 0.18 |
| Bin migration | 0.03 | 0.08 | | 0.01 | | | 0.02 | 0.05 | 0.04 | 0.05 |
| Muon efficiency | 2.0 | 1.9 | 1.7 | 1.6 | 1.6 | 1.5 | 1.6 | 1.5 | 1.5 | 1.5 |
| Mass: PD shape | 0.06 | 0.22 | 0.01 | 0.17 | 0.10 | 0.22 | 0.10 | 0.18 | 0.19 | 0.33 |
| Mass: $\psi(2S)$ offset | 0.03 | 0.17 | 0.02 | 0.15 | 0.02 | 0.02 | | | | 0.01 |
| $p_{\mathrm{T}}^2$: $\chi_c$ feed-down | 0.01 | 0.01 | 0.01 | 0.06 | 0.03 | 0.01 | 0.04 | 0.01 | 0.02 | 0.03 |
| $p_{\mathrm{T}}^2$: $\psi(2S)$ feed-down | 0.12 | 0.04 | 0.01 | 0.09 | 0.01 | 0.02 | 0.02 | 0.02 | 0.03 | 0.04 |
| $p_{\mathrm{T}}^2$: PD shape | 3.5 | 1.8 | 0.20 | 0.4 | 0.30 | 0.11 | 0.01 | 0.7 | 0.32 | 1.2 |
| **Total uncorrelated** | 4.0 | 2.7 | 1.8 | 1.7 | 1.7 | 1.5 | 1.6 | 1.7 | 1.5 | 2.0 |
| **Correlated uncertainties** | | | | | | | | | | |
| GEC efficiency | 0.8 | 0.8 | 0.8 | 0.8 | 0.8 | 0.8 | 0.8 | 0.8 | 0.8 | 0.8 |
| GEC background | 0.8 | 0.8 | 0.8 | 0.8 | 0.8 | 0.8 | 0.8 | 0.8 | 0.8 | 0.8 |
| SPD hits efficiency | 0.03 | 0.03 | 0.03 | 0.03 | 0.03 | 0.03 | 0.03 | 0.03 | 0.03 | 0.03 |
| SPD multiplicity shape | 0.15 | 0.15 | 0.15 | 0.15 | 0.15 | 0.15 | 0.15 | 0.15 | 0.15 | 0.15 |
| **Total correlated** | 1.2 | 1.2 | 1.2 | 1.2 | 1.2 | 1.2 | 1.2 | 1.2 | 1.2 | 1.2 |
| **Total uncertainties** | | | | | | | | | | |
| Systematic (excl. luminosity) | 4.2 | 2.9 | 2.1 | 2.0 | 2.0 | 1.9 | 1.9 | 2.0 | 1.9 | 2.3 |
| Luminosity | 2.9 | 2.9 | 2.9 | 2.9 | 2.9 | 2.9 | 2.9 | 2.9 | 2.9 | 2.9 |
| Statistical | 2.3 | 12.6 | 1.5 | 6.1 | 1.1 | 4.6 | 1.0 | 4.1 | 0.9 | 4.0 |

| $y$ bin Source / state | 3.25–3.5 $J/\psi$ | $\psi(2S)$ | 3.5–3.75 $J/\psi$ | $\psi(2S)$ | 3.75–4.0 $J/\psi$ | $\psi(2S)$ | 4.0–4.25 $J/\psi$ | $\psi(2S)$ | 4.25–4.5 $J/\psi$ | $\psi(2S)$ |
|---|---|---|---|---|---|---|---|---|---|---|
| **Uncorrelated uncertainties** | | | | | | | | | | |
| Simulation sample size | 0.09 | 0.18 | 0.10 | 0.20 | 0.12 | 0.25 | 0.16 | 0.35 | 0.32 | 0.7 |
| Bin migration | 0.01 | 0.02 | 0.02 | 0.02 | | 0.03 | 0.03 | 0.10 | 0.09 | 0.14 |
| Muon efficiency | 1.4 | 1.5 | 1.5 | 1.5 | 1.6 | 1.5 | 1.6 | 1.7 | 1.8 | 2.1 |
| Mass: PD shape | 0.17 | 0.28 | 0.13 | 0.15 | 0.13 | 0.32 | 0.06 | 0.20 | 0.03 | 0.10 |
| Mass: $\psi(2S)$ offset | 0.03 | | 0.03 | 0.02 | 0.02 | 0.02 | 0.06 | 0.02 | 0.04 | 0.09 |
| $p_{\mathrm{T}}^2$: $\chi_c$ feed-down | | 0.01 | 0.02 | | | 0.03 | 0.03 | 0.03 | 0.03 | 0.02 |
| $p_{\mathrm{T}}^2$: $\psi(2S)$ feed-down | 0.04 | 0.01 | 0.01 | 0.02 | 0.01 | 0.04 | 0.05 | 0.01 | | 0.04 |
| $p_{\mathrm{T}}^2$: PD shape | 0.6 | 0.6 | 0.8 | 1.6 | 1.0 | 2.1 | 1.5 | 2.2 | 1.5 | 0.9 |
| **Total uncorrelated** | 1.6 | 1.6 | 1.7 | 2.2 | 1.8 | 2.6 | 2.2 | 2.8 | 2.4 | 2.4 |
| **Correlated uncertainties** | | | | | | | | | | |
| GEC efficiency | 0.8 | 0.8 | 0.8 | 0.8 | 0.8 | 0.8 | 0.8 | 0.8 | 0.8 | 0.8 |
| GEC background | 0.8 | 0.8 | 0.8 | 0.8 | 0.8 | 0.8 | 0.8 | 0.8 | 0.8 | 0.8 |
| SPD hits efficiency | 0.03 | 0.03 | 0.03 | 0.03 | 0.03 | 0.03 | 0.03 | 0.03 | 0.03 | 0.03 |
| SPD multiplicity shape | 0.15 | 0.15 | 0.15 | 0.15 | 0.15 | 0.15 | 0.15 | 0.15 | 0.15 | 0.15 |
| **Total correlated** | 1.2 | 1.2 | 1.2 | 1.2 | 1.2 | 1.2 | 1.2 | 1.2 | 1.2 | 1.2 |
| **Total uncertainties** | | | | | | | | | | |
| Systematic (excl. luminosity) | 1.9 | 2.0 | 2.0 | 2.5 | 2.2 | 2.8 | 2.5 | 3.0 | 2.6 | 2.6 |
| Luminosity | 2.9 | 2.9 | 2.9 | 2.9 | 2.9 | 2.9 | 2.9 | 2.9 | 2.9 | 2.9 |
| Statistical | 0.9 | 3.6 | 1.0 | 4.3 | 1.2 | 5.2 | 1.6 | 7.7 | 2.5 | 16.2 |

Table 2: Systematic and statistical uncertainties per rapidity interval on the exponential slopes $b_{J/\psi}$ and $b_{\psi(2S)}$, in percent. Values below 0.005% are not reported.

| $y$ bin Source / state | 2.0–2.25 $J/\psi$ | $\psi(2S)$ | 2.25–2.5 $J/\psi$ | $\psi(2S)$ | 2.5–2.75 $J/\psi$ | $\psi(2S)$ | 2.75–3.0 $J/\psi$ | $\psi(2S)$ | 3.0–3.25 $J/\psi$ | $\psi(2S)$ |
|---|---|---|---|---|---|---|---|---|---|---|
| Mass: PD | 0.08 | 0.20 | 0.04 | 0.02 | 0.04 | 0.03 | 0.05 | 0.02 | 0.09 | 0.07 |
| Mass: $\psi(2S)$ offset | 0.03 | 0.09 | 0.01 | 0.04 | 0.01 | 0.03 | 0.01 | 0.01 | 0.01 | 0.03 |
| $p_{\mathrm T}^2$: $\chi_{cJ}$ feed-down | 0.05 | 0.05 | 0.03 | | 0.07 | | 0.08 | 0.01 | 0.05 | 0.03 |
| $p_{\mathrm T}^2$: $\psi(2S)$ feed-down | 0.29 | 0.08 | 0.06 | | | | 0.01 | 0.02 | 0.01 | 0.03 |
| $p_{\mathrm T}^2$: PD shape | 4.0 | 0.9 | 0.7 | 0.30 | 0.17 | 2.3 | 0.03 | 1.0 | 0.26 | 0.5 |
| Total systematic | 4.0 | 0.9 | 0.7 | 0.30 | 0.19 | 2.3 | 0.10 | 1.0 | 0.28 | 0.5 |
| Statistical | 2.6 | 14.4 | 1.5 | 8.0 | 1.2 | 5.6 | 1.0 | 5.2 | 0.9 | 5.0 |

| $y$ bin Source / state | 3.25–3.5 $J/\psi$ | $\psi(2S)$ | 3.5–3.75 $J/\psi$ | $\psi(2S)$ | 3.75–4.0 $J/\psi$ | $\psi(2S)$ | 4.0–4.25 $J/\psi$ | $\psi(2S)$ | 4.25–4.5 $J/\psi$ | $\psi(2S)$ |
|---|---|---|---|---|---|---|---|---|---|---|
| Mass: PD | 0.07 | 0.02 | 0.06 | 0.01 | 0.06 | 0.07 | 0.03 | | 0.04 | 0.25 |
| Mass: $\psi(2S)$ offset | 0.01 | 0.01 | | 0.03 | | 0.02 | 0.03 | 0.05 | 0.01 | 0.03 |
| $p_{\mathrm T}^2$: $\chi_{cJ}$ feed-down | 0.04 | 0.01 | 0.07 | 0.02 | 0.04 | | 0.03 | 0.01 | 0.01 | 0.03 |
| $p_{\mathrm T}^2$: $\psi(2S)$ feed-down | 0.08 | | 0.01 | 0.01 | 0.04 | | 0.10 | 0.03 | 0.04 | 0.05 |
| $p_{\mathrm T}^2$: PD shape | 0.17 | 0.26 | 0.13 | 0.6 | 0.08 | 1.8 | 0.4 | 1.7 | 2.2 | 4.1 |
| Total systematic | 0.20 | 0.26 | 0.16 | 0.6 | 0.12 | 1.8 | 0.4 | 1.7 | 2.2 | 4.2 |
| Statistical | 0.9 | 4.6 | 1.0 | 5.4 | 1.2 | 6.3 | 1.7 | 9.9 | 2.9 | 20.4 |

# 6  Results

The signal yields determined by the two-dimensional fit, the efficiencies and the resulting differential $pp \to pJ/\psi p$ and $pp \to p\psi(2S)p$ cross-sections are reported in Tables 4 and 5 in Appendix B. Summing over all rapidity intervals, the total integrated cross-sections for charmonia with $2.0 < y < 4.5$ and muons with $2.0 < \eta < 4.5$ are

$$\sigma_{J/\psi \to \mu^+\mu^-}(2.0 < y_{J/\psi} < 4.5, 2.0 < \eta_{\mu^\pm} < 4.5) = 400 \pm 2 \pm 5 \pm 12 \,\mathrm{pb},$$
$$\sigma_{\psi(2S) \to \mu^+\mu^-}(2.0 < y_{\psi(2S)} < 4.5, 2.0 < \eta_{\mu^\pm} < 4.5) = 9.40 \pm 0.15 \pm 0.13 \pm 0.27 \,\mathrm{pb},$$

where the first uncertainties are statistical, the second systematic and the third are due to the luminosity determination. These values are more precise than those reported in the previous analysis of exclusive $J/\psi$ and $\psi(2S)$ production at $\sqrt{s} = 13\,\mathrm{TeV}$ [34] and are compatible with the previous results at the level of $1.5\,\sigma$.

The measured cross-sections are corrected for the $J/\psi \to \mu^+\mu^-$ and $\psi(2S) \to e^+e^-$ branching fractions [89] and the detector acceptance of the two muons, using simulation. The branching fraction for $\psi(2S) \to e^+e^-$ is used under the assumption of lepton universality as it is more precise than that of $\psi(2S) \to \mu^+\mu^-$. The resulting differential cross-sections are shown in Fig. 7. Theoretical predictions are shown for comparison. The $J/\psi$ cross-section agrees with the NLO prediction [45], which is posterior to the previous 13 TeV measurement [34]. On the other hand, the $\psi(2S)$ cross-section is significantly lower than both the LO and NLO predictions [95], which predate the LHCb measurement. The present result calls for an updated calculation of the differential $\psi(2S)$ prediction.

The $J/\psi$ cross-sections are used to determine the photoproduction cross-section as a function of the photon-proton energy, which is reported in Appendix A.

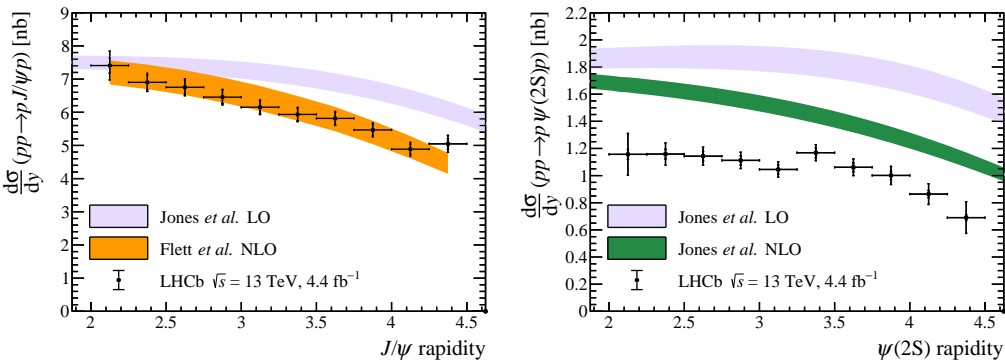

Figure 7: Differential cross-section for (left) $J/\psi$ and (right) $\psi(2S)$ mesons. Theoretical predictions from Jones *et al.* [95, 96] and Flett *et al.* [45] are shown for comparison.

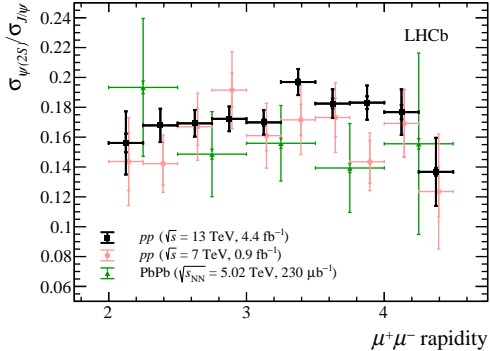

Figure 8: Measured ratio of $\psi(2S)$ and $J/\psi$ cross-sections per rapidity interval. The results from PbPb collisions at $\sqrt{s_{\text{NN}}} = 5.02\,\text{TeV}$ [41] and $pp$ collisions at $\sqrt{s} = 7\,\text{TeV}$ [33] are shown for comparison. The latter data points are slightly offset horizontally to increase visibility.

The ratio of $\psi(2S)$ and $J/\psi$ cross-sections integrated over rapidity is found to be

$$\frac{\sigma_{\psi(2S)}}{\sigma_{J/\psi}} = 0.1763 \pm 0.0029 \pm 0.0008 \pm 0.0039,$$

where the last uncertainty is due to the knowledge of the branching fractions. The luminosity uncertainty cancels in the ratio. The ratio is shown in rapidity intervals in Fig. 8 and agrees with measurements by the LHCb collaboration in $pp$ collisions at $\sqrt{s} = 7\,\text{TeV}$ [33] and PbPb collisions at $\sqrt{s_{\text{NN}}} = 5.02\,\text{TeV}$ [41]. Its average is consistent with those measured by the H1 and ZEUS collaborations [97, 98].

The photoproduction cross-section has an exponential behaviour versus $p_{\text{T}}^2$: $\mathrm{d}\sigma/\mathrm{d}p_{\text{T}}^2 \sim e^{-bp_{\text{T}}^2}$. The exponential slope $b$ can be parameterised as

$$b = b_0 + 4\alpha' \log\left(\frac{W_{\gamma p}}{W_0}\right), \tag{2}$$

where, in Regge theory, $\alpha'$ is the slope of the pomeron trajectory, $W_{\gamma p}$ is the photon-proton centre-of-mass energy defined in Appendix A, $W_0$ is typically taken to be $W_0 = 90\,\text{GeV}$, and $b_0$ is determined experimentally.

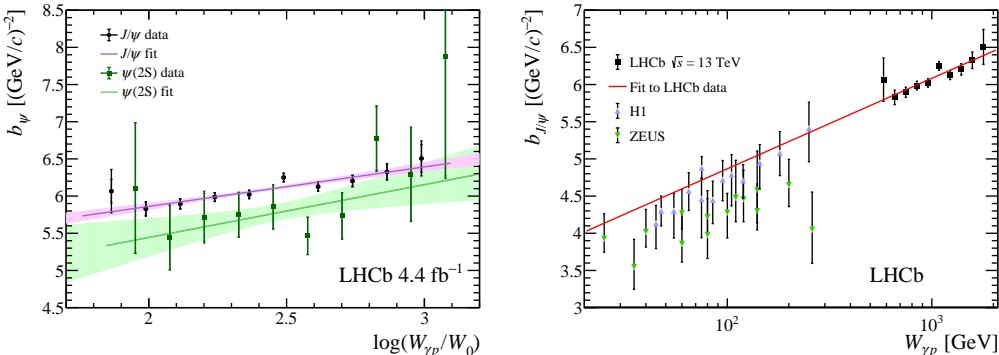

Figure 9: (Left) linear fit to the logarithmic dependence of $b_{J/\psi}$ and $b_{\psi(2S)}$ with respect to the photon-proton energy $W_{\gamma p}$ for $J/\psi$ and $\psi(2S)$ production. The shaded areas represent the 68% C.L. fit uncertainties. (Right) measured $b$ slopes for $J/\psi$ production by the LHCb (this paper), H1 [93, 99] and ZEUS [100] experiments. Superimposed is a line with the slope resulting from the fit to the LHCb data.

A linear fit to the $b$ slopes in intervals of rapidity is shown in Fig. 9. The intercepts and slopes are determined to be

$$\alpha'^{,J/\psi} = 0.133 \pm 0.024 \pm 0.006 \, (\text{GeV}/c)^{-2},$$
$$\alpha'^{,\psi(2S)} = 0.178 \pm 0.124 \pm 0.004 \, (\text{GeV}/c)^{-2},$$
$$b_0^{J/\psi} = 4.80 \pm 0.24 \pm 0.06 \, (\text{GeV}/c)^{-2},$$
$$b_0^{\psi(2S)} = 4.02 \pm 1.23 \pm 0.03 \, (\text{GeV}/c)^{-2},$$

where the first uncertainty is statistical and the second systematic. The systematic uncertainties are obtained by taking the difference of the central value when the fit is performed with and without the systematic uncertainties accounted for in the fit. The fit to $J/\psi$ data agrees with previous determinations in $ep$ collisions [93, 99–101] but is below the prediction of Ref. [102].

## 7 Conclusion

This paper presents the first measurement of the exclusive $\psi(2S)$ cross-section in $pp$ collisions at $\sqrt{s} = 13$ TeV in ten intervals of rapidity between 2.0 and 4.5. The corresponding $J/\psi$ cross-section is updated and the rapidity-dependent ratio of $\psi(2S)$ and $J/\psi$ production is determined for the first time. The results are consistent but more precise than those of Ref. [34]. When expressed as a function of the photon-proton energy, the cross-sections are found to be consistent with previous measurements, but the $\psi(2S)$ cross-section is below theory predictions.

For the first time, the dependence of the $J/\psi$ and $\psi(2S)$ cross-sections on $p_{\text{T}}^2 \sim \Delta t$, where $\Delta t$ is the total transverse momentum transfer, is determined in $pp$ collisions and is found consistent with and more precise than the behaviour observed at HERA [93, 99–101].

## Acknowledgments

We express our gratitude to our colleagues in the CERN accelerator departments for the excellent performance of the LHC. We thank the technical and administrative staff at the LHCb institutes.

**Funding information** We acknowledge support from CERN and from the national agencies: CAPES, CNPq, FAPERJ and FINEP (Brazil); MOST and NSFC (China); CNRS/IN2P3 (France); BMBF, DFG and MPG (Germany); INFN (Italy); NWO (Netherlands); MNiSW and NCN (Poland); MCID/IFA (Romania); MICIU and AEI (Spain); SNSF and SER (Switzerland); NASU (Ukraine); STFC (United Kingdom); DOE NP and NSF (USA). We acknowledge the computing resources that are provided by CERN, IN2P3 (France), KIT and DESY (Germany), INFN (Italy), SURF (Netherlands), PIC (Spain), GridPP (United Kingdom), CSCS (Switzerland), IFIN-HH (Romania), CBPF (Brazil), and Polish WLCG (Poland). We are indebted to the communities behind the multiple open-source software packages on which we depend. Individual groups or members have received support from ARC and ARDC (Australia); Key Research Program of Frontier Sciences of CAS, CAS PIFI, CAS CCEPP, Fundamental Research Funds for the Central Universities, and Sci. & Tech. Program of Guangzhou (China); Minciencias (Colombia); EPLANET, Marie Skłodowska-Curie Actions, ERC and NextGenerationEU (European Union); A*MIDEX, ANR, IPhU and Labex P2IO, and Région Auvergne-Rhône-Alpes (France); AvH Foundation (Germany); ICSC (Italy); Severo Ochoa and María de Maeztu Units of Excellence, GVA, XuntaGal, GENCAT, InTalent-Inditex and Prog. Atracción Talento CM (Spain); SRC (Sweden); the Leverhulme Trust, the Royal Society and UKRI (United Kingdom).

# A Photoproduction cross-section

The differential cross-sections are used to determine the photoproduction cross-section for $J/\psi$ and $\psi(2S)$ mesons. The differential cross-section is factorised into two terms depending on whether the proton travelling from the vertex detector towards the muon chambers interacts electromagnetically (labelled $W_{\gamma p,+}$) or the opposite-direction proton does (labelled $W_{\gamma p,-}$):

$$\frac{\mathrm{d}\sigma}{\mathrm{d}y}(pp \to p\psi p) = S^2(W_{\gamma p,+})\left(k_+ \frac{\mathrm{d}n}{\mathrm{d}k_+}\right)\sigma^{W_{\gamma p,+}}_{\gamma p \to \psi p} + S^2(W_{\gamma p,-})\left(k_- \frac{\mathrm{d}n}{\mathrm{d}k_-}\right)\sigma^{W_{\gamma p,-}}_{\gamma p \to \psi p}, \qquad \text{(A.1)}$$

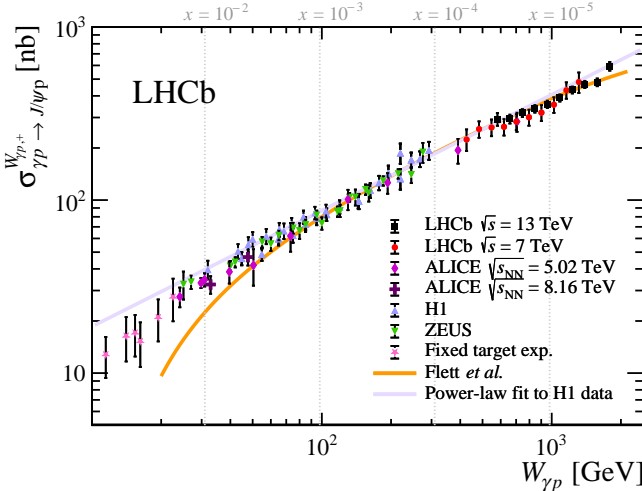

Figure 10: Results for the $J/\psi$ photoproduction cross-section as a function of the photon-proton energy $W_{\gamma p}$ from different experiments listed in Table 3. The LHCb results at $\sqrt{s} = 13$ TeV are estimated with the NLO [45] calculation. Also shown are the NLO theoretical descriptions given by Flett *et al.* [45], as well as a power-law description of the H1 data. The top axis shows the values of $x$ reached for a given photon-proton energy.

Table 3: Previous results used in Fig. 10.

| Marker | Experiment | collision | Energy | Refs. |
|:---:|:---|:---:|:---:|:---:|
| ● | LHCb | $pp$ | $\sqrt{s} = 7\,\text{TeV}$ | [33] |
| ♦ | ALICE | $p\text{Pb}$ | $\sqrt{s_{\text{NN}}} = 5.02\,\text{TeV}$ | [106, 107] |
| + | ALICE | $p\text{Pb}$ | $\sqrt{s_{\text{NN}}} = 8.16\,\text{TeV}$ | [38] |
| ▲ | H1 | $ep$ | $40 < W_{\gamma p} < 305\,\text{GeV}$ | [99] |
| ▲ | H1 | $ep$ | $25 < W_{\gamma p} < 110\,\text{GeV}$ | [93] |
| ▼ | ZEUS | $ep$ | $20 < W_{\gamma p} < 290\,\text{GeV}$ | [100] |
| ✳ | E87 | $\gamma\text{Be}$ | $0 < E_\gamma < 250\,\text{GeV}$ | [108] |
| ✳ | E401 | $\gamma\text{H and }\gamma^2\text{H}$ | $60 < E_\gamma < 300\,\text{GeV}$ | [109] |
| ✳ | E516 | $\gamma\text{H}$ | $60 < E_\gamma < 160\,\text{GeV}$ | [110] |

with $W_{\gamma p,\pm} = \sqrt{M_\psi c^2 \sqrt{s} e^{\pm|y|}}$. The $S^2(W_{\gamma p,\pm})$ terms, the so-called survival factors, are taken from Ref. [103]. The photon flux $\mathrm{d}n/\mathrm{d}k_\pm$ for photons with energy equal to $k_\pm = (M_\psi c^2/2)e^{\pm|y|}$ is calculated following Refs. [104, 105]. The photoproduction cross-sections are given by $\sigma^{W_{\gamma p,\pm}}_{\gamma p \to \psi p}$. The antiparallel $\gamma p$ cross-section, $\sigma^{W_{\gamma p,-}}_{\gamma p \to \psi p}$, corresponds to large values of $x$, as $x \sim M_\psi c^2/\sqrt{s}\, e^{-y}$ [45]. The contribution of this term to Eq. A.1 is therefore expected to be small and can be constrained from theoretical predictions. The antiparallel solution is taken from the $J/\psi$ and $\psi(2S)$ NLO cross-section predictions from Refs. [45, 95] and subtracted. Figure 10 shows the measured photoproduction cross-section for $J/\psi$ mesons and compares it with previous measurements listed in Table 3. The numerical values are listed in Table 8.

The LHCb $J/\psi$ data at $\sqrt{s} = 13\,\text{TeV}$ are in agreement with the NLO description [45]. They also follow a power-law fit to H1 data [93].

# B Numerical results

Tables 4 and 5 present the differential cross-sections in rapidity bins, shown in Fig. 7, and the breakdown of uncertainties for $J/\psi \to \mu^+\mu^-$ and $\psi(2S) \to \mu^+\mu^-$, respectively. Their ratio is given in Table 6. Table 7 lists the exponential slopes for $J/\psi$ and $\psi(2S)$ in each rapidity bin, corresponding to Fig. 9. Table 8 lists the values entering the computation of the parallel cross-sections $\sigma^{W_{\gamma p,+}}_{\gamma p \to J/\psi p}$ displayed in Fig. 10.

# C Fits in all rapidity bins

The distributions of dimuon mass and $p_\text{T}^2$ in each rapidity interval are shown in Fig. 11. The results of the two-dimensional fits described in the main text are overlaid.

Table 4: Differential CEP $J/\psi \to \mu^+\mu^-$ yields and cross-sections corrected for efficiency ($\epsilon_{\text{tot}}$), acceptance ($\epsilon_{\text{Geom.Acc.}}$) and branching fraction. The systematic uncertainties are split between those uncorrelated across $y$ ranges, those that are 100% correlated and the luminosity uncertainty.

| $y_{J/\psi}$ bin | 2.0–2.25 | 2.25–2.5 | 2.5–2.75 | 2.75–3.0 | 3.0–3.25 |
|---|---|---|---|---|---|
| $N_{\text{sig}}$ | $4998 \pm 113$ | $18095 \pm 265$ | $31591 \pm 361$ | $41640 \pm 402$ | $47690 \pm 432$ |
| $\epsilon_{\text{tot}}$ | $0.313 \pm 0.028$ | $0.403 \pm 0.030$ | $0.443 \pm 0.030$ | $0.456 \pm 0.029$ | $0.467 \pm 0.027$ |
| $\epsilon_{\text{Geom.Acc.}}$ | $0.095 \pm 0.002$ | $0.287 \pm 0.003$ | $0.466 \pm 0.003$ | $0.623 \pm 0.003$ | $0.732 \pm 0.003$ |
| $\mathrm{d}\sigma/\mathrm{d}y$ [nb] | 7.41 | 6.90 | 6.75 | 6.46 | 6.15 |
| Stat. unc. [nb] | 0.17 | 0.10 | 0.08 | 0.06 | 0.06 |
| Uncorr. syst. unc. [nb] | 0.30 | 0.12 | 0.11 | 0.10 | 0.09 |
| Corr. syst. unc. [nb] | 0.23 | 0.21 | 0.21 | 0.20 | 0.19 |
| Lumi. unc. [nb] | 0.21 | 0.20 | 0.20 | 0.19 | 0.18 |
| $y_{J/\psi}$ bin | 3.25–3.5 | 3.5–3.75 | 3.75–4.0 | 4.0–4.25 | 4.25–4.5 |
| $N_{\text{sig}}$ | $47303 \pm 436$ | $39878 \pm 394$ | $26727 \pm 329$ | $14428 \pm 236$ | $4349 \pm 108$ |
| $\epsilon_{\text{tot}}$ | $0.479 \pm 0.027$ | $0.484 \pm 0.028$ | $0.462 \pm 0.029$ | $0.435 \pm 0.029$ | $0.399 \pm 0.029$ |
| $\epsilon_{\text{Geom.Acc.}}$ | $0.733 \pm 0.003$ | $0.625 \pm 0.003$ | $0.467 \pm 0.003$ | $0.300 \pm 0.003$ | $0.095 \pm 0.002$ |
| $\mathrm{d}\sigma/\mathrm{d}y$ [nb] | 5.93 | 5.82 | 5.47 | 4.89 | 5.05 |
| Stat. unc. [nb] | 0.05 | 0.06 | 0.07 | 0.08 | 0.13 |
| Uncorr. syst. unc. [nb] | 0.09 | 0.10 | 0.10 | 0.11 | 0.12 |
| Corr. syst. unc. [nb] | 0.18 | 0.18 | 0.17 | 0.15 | 0.16 |
| Lumi. unc. [nb] | 0.17 | 0.17 | 0.16 | 0.14 | 0.15 |

Table 5: Differential CEP $\psi(2S) \to \mu^+\mu^-$ yields and cross-sections corrected for efficiency ($\epsilon_{\text{tot}}$), acceptance ($\epsilon_{\text{Geom.Acc.}}$) and branching fraction. The systematic uncertainties are split between those uncorrelated across $y$ ranges, those that are 100% correlated and the luminosity uncertainty.

| $y_{\psi(2S)}$ bin | 2.0–2.25 | 2.25–2.5 | 2.5–2.75 | 2.75–3.0 | 3.0–3.25 |
|---|---|---|---|---|---|
| $N_{\text{sig}}$ | $127 \pm 16$ | $491 \pm 30$ | $845 \pm 39$ | $1088 \pm 44$ | $1163 \pm 47$ |
| $\epsilon_{\text{tot}}$ | $0.400 \pm 0.032$ | $0.494 \pm 0.032$ | $0.527 \pm 0.032$ | $0.518 \pm 0.032$ | $0.502 \pm 0.031$ |
| $\epsilon_{\text{Geom.Acc.}}$ | $0.091 \pm 0.002$ | $0.284 \pm 0.003$ | $0.465 \pm 0.003$ | $0.626 \pm 0.003$ | $0.735 \pm 0.003$ |
| $\mathrm{d}\sigma/\mathrm{d}y$ [nb] | 1.16 | 1.16 | 1.14 | 1.11 | 1.05 |
| Stat. unc. [nb] | 0.15 | 0.07 | 0.05 | 0.05 | 0.04 |
| Uncorr. syst. unc. [nb] | 0.03 | 0.02 | 0.02 | 0.02 | 0.02 |
| Corr. syst. unc. [nb] | 0.04 | 0.04 | 0.04 | 0.03 | 0.03 |
| Lumi. unc. [nb] | 0.03 | 0.03 | 0.03 | 0.03 | 0.03 |
| $y_{\psi(2S)}$ bin | 3.25–3.5 | 3.5–3.75 | 3.75–4.0 | 4.0–4.25 | 4.25–4.5 |
| $N_{\text{sig}}$ | $1319 \pm 48$ | $980 \pm 42$ | $648 \pm 34$ | $323 \pm 25$ | $81 \pm 13$ |
| $\epsilon_{\text{tot}}$ | $0.505 \pm 0.030$ | $0.492 \pm 0.030$ | $0.457 \pm 0.030$ | $0.427 \pm 0.030$ | $0.394 \pm 0.035$ |
| $\epsilon_{\text{Geom.Acc.}}$ | $0.740 \pm 0.003$ | $0.622 \pm 0.003$ | $0.469 \pm 0.003$ | $0.290 \pm 0.003$ | $0.099 \pm 0.002$ |
| $\mathrm{d}\sigma/\mathrm{d}y$ [nb] | 1.17 | 1.06 | 1.00 | 0.86 | 0.69 |
| Stat. unc. [nb] | 0.04 | 0.05 | 0.05 | 0.07 | 0.11 |
| Uncorr. syst. unc. [nb] | 0.02 | 0.02 | 0.03 | 0.02 | 0.02 |
| Corr. syst. unc. [nb] | 0.04 | 0.03 | 0.03 | 0.03 | 0.02 |
| Lumi. unc. [nb] | 0.03 | 0.03 | 0.03 | 0.02 | 0.02 |

Table 6: Ratio of the CEP $J/\psi \to \mu^+\mu^-$ and $\psi(2S) \to \mu^+\mu^-$ cross-sections per rapidity bin.

| $y$ bin | 2.0–2.25 | 2.25–2.5 | 2.5–2.75 | 2.75–3.0 | 3.0–3.25 |
|---|---|---|---|---|---|
| $\frac{\mathrm{d}\sigma_{\psi(2S)}/\mathrm{d}y}{\mathrm{d}\sigma_{J/\psi}/\mathrm{d}y}$ | 0.156 | 0.168 | 0.169 | 0.172 | 0.170 |
| Stat. unc. | 0.020 | 0.010 | 0.008 | 0.007 | 0.007 |
| Syst. unc. | 0.006 | 0.001 | 0.001 | 0.001 | 0.002 |
| BF unc. | 0.003 | 0.004 | 0.004 | 0.004 | 0.004 |
| $y$ bin | 3.25–3.5 | 3.5–3.75 | 3.75–4.0 | 4.0–4.25 | 4.25–4.5 |
| $\frac{\mathrm{d}\sigma_{\psi(2S)}/\mathrm{d}y}{\mathrm{d}\sigma_{J/\psi}/\mathrm{d}y}$ | 0.197 | 0.182 | 0.183 | 0.177 | 0.137 |
| Stat. unc. | 0.007 | 0.008 | 0.010 | 0.014 | 0.022 |
| Syst. unc. | 0.002 | 0.003 | 0.004 | 0.005 | 0.003 |
| BF unc. | 0.004 | 0.004 | 0.004 | 0.004 | 0.003 |

Table 7: Values of the $b$ slopes measured in the fit.

| $y$ | $\log\left(\frac{W_{\gamma p}^{J/\psi}}{W_0}\right)$ | $b^{J/\psi}$ | $\log\left(\frac{W_{\gamma p}^{\psi(2S)}}{W_0}\right)$ | $b^{\psi(2S)}$ |
|---|---|---|---|---|
| 2.0–2.25 | 1.86 | $6.07 \pm 0.16 \pm 0.25$ | 1.95 | $6.1 \pm 0.9 \pm 0.1$ |
| 2.25–2.5 | 1.99 | $5.83 \pm 0.09 \pm 0.04$ | 2.08 | $5.4 \pm 0.4 \pm 0.0$ |
| 2.5–2.75 | 2.11 | $5.90 \pm 0.07 \pm 0.01$ | 2.20 | $5.7 \pm 0.3 \pm 0.1$ |
| 2.75–3.0 | 2.24 | $5.99 \pm 0.06 \pm 0.01$ | 2.33 | $5.7 \pm 0.3 \pm 0.1$ |
| 3.0–3.25 | 2.36 | $6.02 \pm 0.05 \pm 0.02$ | 2.45 | $5.9 \pm 0.3 \pm 0.0$ |
| 3.25–3.5 | 2.49 | $6.25 \pm 0.06 \pm 0.01$ | 2.58 | $5.5 \pm 0.2 \pm 0.0$ |
| 3.5–3.75 | 2.61 | $6.13 \pm 0.06 \pm 0.01$ | 2.70 | $5.7 \pm 0.3 \pm 0.0$ |
| 3.75–4.0 | 2.74 | $6.20 \pm 0.08 \pm 0.01$ | 2.83 | $6.8 \pm 0.4 \pm 0.1$ |
| 4.0–4.25 | 2.86 | $6.33 \pm 0.11 \pm 0.02$ | 2.95 | $6.3 \pm 0.6 \pm 0.1$ |
| 4.25–4.5 | 2.99 | $6.51 \pm 0.19 \pm 0.14$ | 3.08 | $7.9 \pm 1.6 \pm 0.3$ |

Table 8: Values of the differential cross-section, survival factor, photon flux and antiparallel photoproduction cross-section used for the calculation of the parallel cross-section. The antiparallel $\gamma p$ value is taken from the FMRT NLO description [45].

| $y_{J/\psi}$ **bin** | **2.0–2.25** | **2.25–2.5** | **2.5–2.75** | **2.75–3.0** | **3.0–3.25** |
|---|---|---|---|---|---|
| $\mathbf{d}\sigma/\mathbf{d}y$ [nb] | $7.41 \pm 0.43$ | $6.90 \pm 0.27$ | $6.75 \pm 0.25$ | $6.46 \pm 0.24$ | $6.15 \pm 0.22$ |
| $S^2(W_{\gamma p,+})$ | 0.786 | 0.774 | 0.762 | 0.748 | 0.732 |
| $k_+\frac{\mathrm{d}n}{\mathrm{d}k_+}(\times 10^{-3})$ | 22.7 | 21.6 | 20.4 | 19.2 | 18.0 |
| $S^2(W_{\gamma p,-})$ | 0.885 | 0.888 | 0.891 | 0.893 | 0.896 |
| $k_-\frac{\mathrm{d}n}{\mathrm{d}k_-}(\times 10^{-3})$ | 42.5 | 43.7 | 44.9 | 46.0 | 47.2 |
| $\sigma^{W_{\gamma p,-}}_{\gamma p \to J/\psi p}$ [nb] | $57.7 \pm 2.7$ | $51.1 \pm 2.3$ | $45.0 \pm 2.0$ | $39.2 \pm 1.7$ | $33.9 \pm 1.4$ |
| $\sigma^{W_{\gamma p,+}}_{\gamma p \to J/\psi p}$ [nb] | $294 \pm 25$ | $294 \pm 17$ | $319 \pm 17$ | $337 \pm 17$ | $358 \pm 17$ |
| $y_{J/\psi}$ **bin** | **3.25–3.5** | **3.5–3.75** | **3.75–4.0** | **4.0–4.25** | **4.25–4.5** |
| $\mathbf{d}\sigma/\mathbf{d}y$ [nb] | $5.93 \pm 0.21$ | $5.82 \pm 0.21$ | $5.47 \pm 0.21$ | $4.89 \pm 0.21$ | $5.05 \pm 0.26$ |
| $S^2(W_{\gamma p,+})$ | 0.715 | 0.695 | 0.672 | 0.647 | 0.618 |
| $k_+\frac{\mathrm{d}n}{\mathrm{d}k_+}(\times 10^{-3})$ | 16.8 | 21.6 | 14.5 | 13.3 | 12.1 |
| $S^2(W_{\gamma p,-})$ | 0.899 | 0.901 | 0.903 | 0.905 | 0.907 |
| $k_-\frac{\mathrm{d}n}{\mathrm{d}k_-}(\times 10^{-3})$ | 48.3 | 49.5 | 50.7 | 51.8 | 53.0 |
| $\sigma^{W_{\gamma p,-}}_{\gamma p \to J/\psi p}$ [nb] | $29.0 \pm 1.2$ | $24.4 \pm 0.9$ | $20.1 \pm 0.7$ | $16.2 \pm 0.6$ | $12.6 \pm 0.4$ |
| $\sigma^{W_{\gamma p,+}}_{\gamma p \to J/\psi p}$ [nb] | $389 \pm 18$ | $433 \pm 20$ | $467 \pm 22$ | $480 \pm 25$ | $594 \pm 34$ |

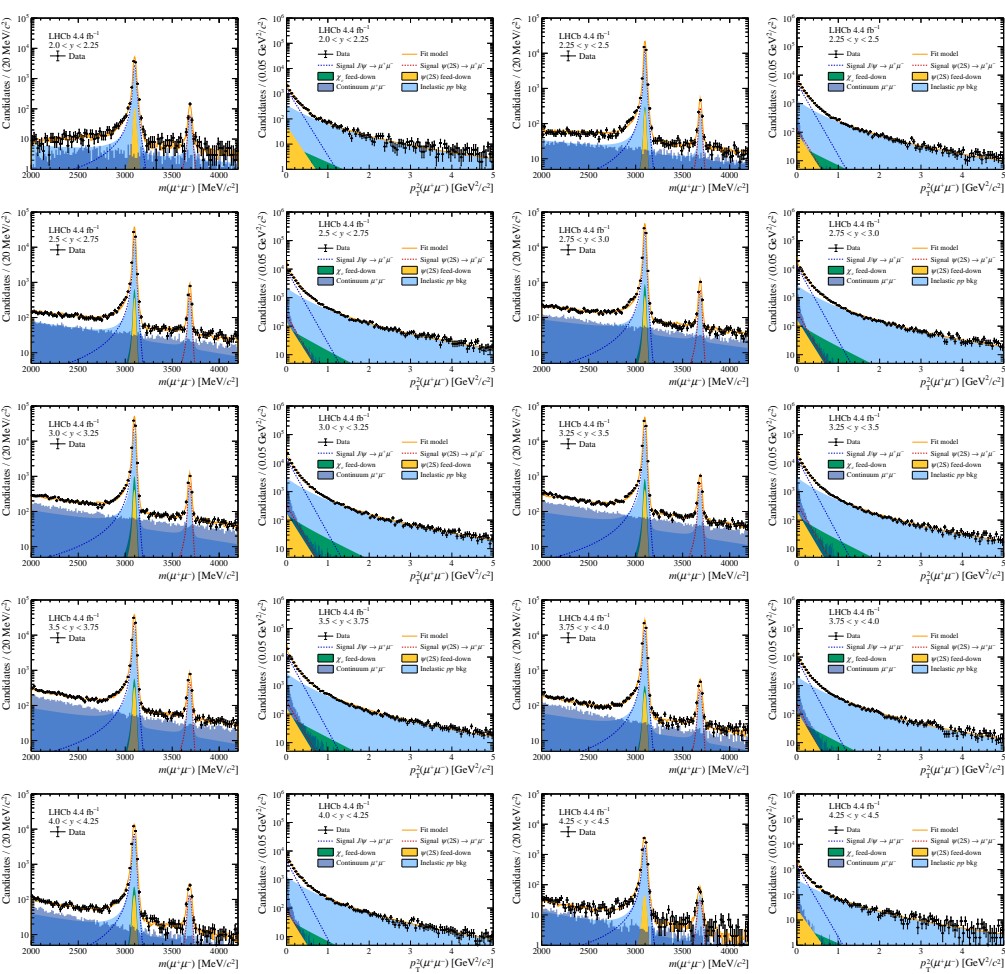

Figure 11: Distributions of dimuon mass and $p_{\mathrm{T}}^2$ for the CEP sample in different regions of rapidity. The results of the two-dimensional fits are overlaid.

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

# LHCb collaboration

R. Aaij[36], A.S.W. Abdelmotteleb[55], C. Abellan Beteta[49], F. Abudinén[55], T. Ackernley[59], A. A. Adefisoye[67], B. Adeva[45], M. Adinolfi[53], P. Adlarson[79], C. Agapopoulou[13], C.A. Aidala[80], Z. Ajaltouni[11], S. Akar[64], K. Akiba[36], P. Albicocco[26], J. Albrecht[18], F. Alessio[47], M. Alexander[58], Z. Aliouche[61], P. Alvarez Cartelle[54], R. Amalric[15], S. Amato[3], J.L. Amey[53], Y. Amhis[13,47], L. An[6], L. Anderlini[25], M. Andersson[49], A. Andreianov[42], P. Andreola[49],



M. Andreotti[24], D. Andreou[67], A. Anelli[29,p], D. Ao[7], F. Archilli[35,v], M. Argenton[24], S. Arguedas Cuendis[9], A. Artamonov[42], M. Artuso[67], E. Aslanides[12], R. Ataíde Da Silva[48], M. Atzeni[63], B. Audurier[14], D. Bacher[62], I. Bachiller Perea[10], S. Bachmann[20], M. Bachmayer[48], J.J. Back[55], P. Baladron Rodriguez[45], V. Balagura[14], W. Baldini[24], H. Bao[7], J. Baptista de Souza Leite[59], M. Barbetti[25,m], I. R. Barbosa[68], R.J. Barlow[61], M. Barnyakov[23], S. Barsuk[13], W. Barter[57], M. Bartolini[54], J. Bartz[67], J.M. Basels[16], G. Bassi[33,s], B. Batsukh[5], A. Bay[48], A. Beck[55], M. Becker[18], F. Bedeschi[33], I.B. Bediaga[2], S. Belin[45], V. Bellee[49], K. Belous[42], I. Belov[27], I. Belyaev[34], G. Benane[12], G. Bencivenni[26], E. Ben-Haim[15], A. Berezhnoy[42], R. Bernet[49], S. Bernet Andres[43], A. Bertolin[31], C. Betancourt[49], F. Betti[57], J. Bex[54], Ia. Bezshyiko[49], J. Bhom[39], M.S. Bieker[18], N.V. Biesuz[24], P. Billoir[15], A. Biolchini[36], M. Birch[60], F.C.R. Bishop[10], A. Bitadze[61], A. Bizzeti, T. Blake[55], F. Blanc[48], J.E. Blank[18], S. Blusk[67], V. Bocharnikov[42], J.A. Boelhauve[18], O. Boente Garcia[14], T. Boettcher[64], A. Bohare[57], A. Boldyrev[42], C.S. Bolognani[76], R. Bolzonella[24,l], N. Bondar[42], F. Borgato[31,q], S. Borghi[61], M. Borsato[29,p], J.T. Borsuk[39], S.A. Bouchiba[48], T.J.V. Bowcock[59], A. Boyer[47], C. Bozzi[24], P. Braat[36], A. Brea Rodriguez[48], N. Breer[18], J. Brodzicka[39], A. Brossa Gonzalo[45,55,44,†], J. Brown[59], D. Brundu[30], E. Buchanan[57], A. Buonaura[49], L. Buonincontri[31,q], A.T. Burke[61], C. Burr[47], A. Butkevich[42], J.S. Butter[54], J. Buytaert[47], W. Byczynski[47], S. Cadeddu[30], H. Cai[72], R. Calabrese[24,l], S. Calderon Ramirez[9], L. Calefice[44], S. Cali[26], M. Calvi[29,p], M. Calvo Gomez[43], P. Camargo Magalhaes[2,z], J. I. Cambon Bouzas[45], P. Campana[26], D.H. Campora Perez[76], A.F. Campoverde Quezada[7], S. Capelli[29], L. Capriotti[24], R. Caravaca-Mora[9], A. Carbone[23,j], L. Carcedo Salgado[45], R. Cardinale[27,n], A. Cardini[30], P. Carniti[29,p], L. Carus[20], A. Casais Vidal[63], R. Caspary[20], G. Casse[59], J. Castro Godinez[9], M. Cattaneo[47], G. Cavallero[24,47], V. Cavallini[24,l], S. Celani[20], D. Cervenkov[62], S. Cesare[28,o], A.J. Chadwick[59], I. Chahrour[80], M. Charles[15], Ph. Charpentier[47], E. Chatzianagnostou[36], C.A. Chavez Barajas[59], M. Chefdeville[10], C. Chen[12], S. Chen[5], Z. Chen[7], A. Chernov[39], S. Chernyshenko[51], V. Chobanova[78], S. Cholak[48], M. Chrzaszcz[39], A. Chubykin[42], V. Chulikov[42], P. Ciambrone[26], X. Cid Vidal[45], G. Ciezarek[47], P. Cifra[47], P.E.L. Clarke[57], M. Clemencic[47], H.V. Cliff[54], J. Closier[47], C. Cocha Toapaxi[20], V. Coco[47], J. Cogan[12], E. Cogneras[11], L. Cojocariu[41], P. Collins[47], T. Colombo[47], A. Comerma-Montells[44], L. Congedo[22], A. Contu[30], N. Cooke[58], I. Corredoira[45], A. Correia[15], G. Corti[47], J.J. Cottee Meldrum[53], B. Couturier[47], D.C. Craik[49], M. Cruz Torres[2,g], E. Curras Rivera[48], R. Currie[57], C.L. Da Silva[66], S. Dadabaev[42], L. Dai[69], X. Dai[6], E. Dall'Occo[18], J. Dalseno[45], C. D'Ambrosio[47], J. Daniel[11], A. Danilina[42], P. d'Argent[22], A. Davidson[55], J.E. Davies[61], A. Davis[61], O. De Aguiar Francisco[61], C. De Angelis[30,k], F. De Benedetti[47], J. de Boer[36], K. De Bruyn[75], S. De Capua[61], M. De Cian[20,47], U. De Freitas Carneiro Da Graca[2,b], E. De Lucia[26], J.M. De Miranda[2], L. De Paula[3], M. De Serio[22,h], P. De Simone[26], F. De Vellis[18], J.A. de Vries[76], F. Debernardis[22], D. Decamp[10], V. Dedu[12], L. Del Buono[15], B. Delaney[63], H.-P. Dembinski[18], J. Deng[8], V. Denysenko[49], O. Deschamps[11], F. Dettori[30,k], B. Dey[74], P. Di Nezza[26], I. Diachkov[42], S. Didenko[42], S. Ding[67], L. Dittmann[20], V. Dobishuk[51], A. D. Docheva[58], C. Dong[4], A.M. Donohoe[21], F. Dordei[30], A.C. dos Reis[2], A. D. Dowling[67], W. Duan[70], P. Duda[77], M.W. Dudek[39], L. Dufour[47], V. Duk[32], P. Durante[47], M. M. Duras[77], J.M. Durham[66], O. D. Durmus[74], A. Dziurda[39], A. Dzyuba[42], S. Easo[56], E. Eckstein[17], U. Egede[1], A. Egorychev[42], V. Egorychev[42], S. Eisenhardt[57], E. Ejopu[61], L. Eklund[79], M. Elashri[64], J. Ellbracht[18], S. Ely[60], A. Ene[41], E. Epple[64], J. Eschle[67], S. Esen[20], T. Evans[61], F. Fabiano[30,k], L.N. Falcao[2], Y. Fan[7], B. Fang[72], L. Fantini[32,r,47], M. Faria[48], K. Farmer[57], D. Fazzini[29,p], L. Felkowski[77], M. Feng[5,7], M. Feo[18,47], M. Fernandez Gomez[45], A.D. Fernez[65], F. Ferrari[23], F. Ferreira Rodrigues[3], M. Ferrillo[49], M. Ferro-Luzzi[47], S. Filippov[42], R.A. Fini[22], M. Fiorini[24,l], K.L. Fischer[62], D.S. Fitzgerald[80], C. Fitzpatrick[61], F. Fleuret[14], M. Fontana[23], L. F. Foreman[61], R. Forty[47], D. Foulds-Holt[54], M. Franco Sevilla[65], M. Frank[47], E. Franzoso[24,l], G. Frau[61], C. Frei[47], D.A. Friday[61], J. Fu[7], Q. Fuehring[18], Y. Fujii[1], T. Fulghesu[15], E. Gabriel[36], G. Galati[22], M.D. Galati[36], A. Gallas Torreira[45], D. Galli[23,j], S. Gambetta[57], M. Gandelman[3], P. Gandini[28], B. Ganie[61], H. Gao[7], R. Gao[62], Y. Gao[8], Y. Gao[6], Y. Gao[8], M. Garau[30,k], L.M. Garcia Martin[48], P. Garcia Moreno[44], J. García Pardiñas[47], K. G. Garg[8], L. Garrido[44], C. Gaspar[47], R.E. Geertsema[36], L.L. Gerken[18], E. Gersabeck[61], M. Gersabeck[61], T. Gershon[55],

Z. Ghorbanimoghaddam[53], L. Giambastiani[31,q], F. I. Giasemis[15,e], V. Gibson[54], H.K. Giemza[40], A.L. Gilman[62], M. Giovannetti[26], A. Gioventù[44], P. Gironella Gironell[44], C. Giugliano[24,l], M.A. Giza[39], E.L. Gkougkousis[60], F.C. Glaser[13,20], V.V. Gligorov[15,47], C. Göbel[68], E. Golobardes[43], D. Golubkov[42], A. Golutvin[60,42,47], A. Gomes[2,a,†], S. Gomez Fernandez[44], F. Goncalves Abrantes[62], M. Goncerz[39], G. Gong[4], J. A. Gooding[18], I.V. Gorelov[42], C. Gotti[29], J.P. Grabowski[17], L.A. Granado Cardoso[47], E. Graugés[44], E. Graverini[48,t], L. Grazette[55], G. Graziani, A. T. Grecu[41], L.M. Greeven[36], N.A. Grieser[64], L. Grillo[58], S. Gromov[42], C. Gu[14], M. Guarise[24], M. Guittiere[13], V. Guliaeva[42], P. A. Günther[20], A.-K. Guseinov[48], E. Gushchin[42], Y. Guz[6,42,47], T. Gys[47], K. Habermann[17], T. Hadavizadeh[1], C. Hadjivasiliou[65], G. Haefeli[48], C. Haen[47], J. Haimberger[47], M. Hajheidari[47], M.M. Halvorsen[47], P.M. Hamilton[65], J. Hammerich[59], Q. Han[8], X. Han[20], S. Hansmann-Menzemer[20], L. Hao[7], N. Harnew[62], M. Hartmann[13], J. He[7,c], F. Hemmer[47], C. Henderson[64], R.D.L. Henderson[1,55], A.M. Hennequin[47], K. Hennessy[59], L. Henry[48], J. Herd[60], P. Herrero Gascon[20], J. Heuel[16], A. Hicheur[3], G. Hijano Mendizabal[49], D. Hill[48], S.E. Hollitt[18], J. Horswill[61], R. Hou[8], Y. Hou[11], N. Howarth[59], J. Hu[20], J. Hu[70], W. Hu[6], X. Hu[4], W. Huang[7], W. Hulsbergen[36], R.J. Hunter[55], M. Hushchyn[42], D. Hutchcroft[59], D. Ilin[42], P. Ilten[64], A. Inglessi[42], A. Iniukhin[42], A. Ishteev[42], K. Ivshin[42], R. Jacobsson[47], H. Jage[16], S.J. Jaimes Elles[46,73], S. Jakobsen[47], E. Jans[36], B.K. Jashal[46], A. Jawahery[65,47], V. Jevtic[18], E. Jiang[65], X. Jiang[5,7], Y. Jiang[7], Y. J. Jiang[6], M. John[62], D. Johnson[52], C.R. Jones[54], T.P. Jones[55], S. Joshi[40], B. Jost[47], N. Jurik[47], I. Juszczak[39], D. Kaminaris[48], S. Kandybei[50], M. Kane[57], Y. Kang[4], C. Kar[11], M. Karacson[47], D. Karpenkov[42], A. Kauniskangas[48], J.W. Kautz[64], F. Keizer[47], M. Kenzie[54], T. Ketel[36], B. Khanji[67], A. Kharisova[42], S. Kholodenko[33,47], G. Khreich[13], T. Kirn[16], V.S. Kirsebom[29,p], O. Kitouni[63], S. Klaver[37], N. Kleijne[33,s], K. Klimaszewski[40], M.R. Kmiec[40], S. Koliiev[51], L. Kolk[18], A. Konoplyannikov[42], P. Kopciewicz[38,47], P. Koppenburg[36], M. Korolev[42], I. Kostiuk[36], O. Kot[51], S. Kotriakhova, A. Kozachuk[42], P. Kravchenko[42], L. Kravchuk[42], M. Kreps[55], P. Krokovny[42], W. Krupa[67], W. Krzemien[40], O.K. Kshyvanskyi[51], J. Kubat[20], S. Kubis[77], M. Kucharczyk[39], V. Kudryavtsev[42], E. Kulikova[42], A. Kupsc[79], B. K. Kutsenko[12], D. Lacarrere[47], A. Lai[30], A. Lampis[30], D. Lancierini[54], C. Landesa Gomez[45], J.J. Lane[1], R. Lane[53], C. Langenbruch[20], J. Langer[18], O. Lantwin[42], T. Latham[55], F. Lazzari[33,t], C. Lazzeroni[52], R. Le Gac[12], R. Lefèvre[11], A. Leflat[42], S. Legotin[42], M. Lehuraux[55], E. Lemos Cid[47], O. Leroy[12], T. Lesiak[39], B. Leverington[20], A. Li[4], H. Li[70], K. Li[8], L. Li[61], P. Li[47], P.-R. Li[71], Q. Li[5,7], S. Li[8], T. Li[5,d], T. Li[70], Y. Li[8], Y. Li[5], Z. Lian[4], X. Liang[67], S. Libralon[46], C. Lin[7], T. Lin[56], R. Lindner[47], V. Lisovskyi[48], R. Litvinov[30,47], F. L. Liu[1], G. Liu[70], K. Liu[71], S. Liu[5,7], Y. Liu[57], Y. Liu[71], Y. L. Liu[60], A. Lobo Salvia[44], A. Loi[30], J. Lomba Castro[45], T. Long[54], J.H. Lopes[3], A. Lopez Huertas[44], S. López Soliño[45], A. Loya Villalpando[36], C. Lucarelli[25,m], D. Lucchesi[31,q], M. Lucio Martinez[76], V. Lukashenko[36,51], Y. Luo[6], A. Lupato[31], E. Luppi[24,l], K. Lynch[21], X.-R. Lyu[7], G. M. Ma[4], R. Ma[7], S. Maccolini[18], F. Machefert[13], F. Maciuc[41], B. Mack[67], I. Mackay[62], L. M. Mackey[67], L.R. Madhan Mohan[54], M. J. Madurai[52], A. Maevskiy[42], D. Magdalinski[36], D. Maisuzenko[42], M.W. Majewski[38], J.J. Malczewski[39], S. Malde[62], L. Malentacca[47], A. Malinin[42], T. Maltsev[42], G. Manca[30,k], G. Mancinelli[12], C. Mancuso[28,13,o], R. Manera Escalero[44], D. Manuzzi[23], D. Marangotto[28,o], J.F. Marchand[10], R. Marchevski[48], U. Marconi[23], S. Mariani[47], C. Marin Benito[44], J. Marks[20], A.M. Marshall[53], G. Martelli[32,r], G. Martellotti[34], L. Martinazzoli[47], M. Martinelli[29,p], D. Martinez Santos[45], F. Martinez Vidal[46], A. Massafferri[2], R. Matev[47], A. Mathad[47], V. Matiunin[42], C. Matteuzzi[67], K.R. Mattioli[14], A. Mauri[60], E. Maurice[14], J. Mauricio[44], P. Mayencourt[48], M. Mazurek[40], M. McCann[60], L. Mcconnell[21], T.H. McGrath[61], N.T. McHugh[58], A. McNab[61], R. McNulty[21], B. Meadows[64], G. Meier[18], D. Melnychuk[40], F. M. Meng[4], M. Merk[36,76], A. Merli[48], L. Meyer Garcia[65], D. Miao[5,7], H. Miao[7], M. Mikhasenko[17,f], D.A. Milanes[73], A. Minotti[29,p], E. Minucci[67], T. Miralles[11], B. Mitreska[18], D.S. Mitzel[18], A. Modak[56], A. Mödden[18], R.A. Mohammed[62], R.D. Moise[16], S. Mokhnenko[42], T. Mombächer[47], M. Monk[55,1], S. Monteil[11], A. Morcillo Gomez[45], G. Morello[26], M.J. Morello[33,s], M.P. Morgenthaler[20], A.B. Morris[47], A.G. Morris[12], R. Mountain[67], H. Mu[4], Z. M. Mu[6], E. Muhammad[55], F. Muheim[57], M. Mulder[75], K. Müller[49], F. Muñoz-Rojas[9], R. Murta[60], P. Naik[59], T. Nakada[48], R. Nandakumar[56], T. Nanut[47], I. Nasteva[3], M. Needham[57], N. Neri[28,o], S. Neubert[17], N. Neufeld[47], P. Neustroev[42], J. Nicolini[18,13], D. Nicotra[76], E.M. Niel[48],

N. Nikitin[42], P. Nogarolli[3], P. Nogga[17], N.S. Nolte[63], C. Normand[53], J. Novoa Fernandez[45], G. Nowak[64], C. Nunez[80], H. N. Nur[58], A. Oblakowska-Mucha[38], V. Obraztsov[42], T. Oeser[16], S. Okamura[24,l], A. Okhotnikov[42], O. Okhrimenko[51], R. Oldeman[30,k], F. Oliva[57], M. Olocco[18], C.J.G. Onderwater[76], R.H. O'Neil[57], J.M. Otalora Goicochea[3], P. Owen[49], A. Oyanguren[46], O. Ozcelik[57], A. Padee[40], K.O. Padeken[17], B. Pagare[55], P.R. Pais[20], T. Pajero[47], A. Palano[22], M. Palutan[26], G. Panshin[42], L. Paolucci[55], A. Papanestis[56], M. Pappagallo[22,h], L.L. Pappalardo[24,l], C. Pappenheimer[64], C. Parkes[61], B. Passalacqua[24], G. Passaleva[25], D. Passaro[33,s], A. Pastore[22], M. Patel[60], J. Patoc[62], C. Patrignani[23,j], A. Paul[67], C.J. Pawley[76], A. Pellegrino[36], J. Peng[5,7], M. Pepe Altarelli[26], S. Perazzini[23], D. Pereima[42], H. Pereira Da Costa[66], A. Pereiro Castro[45], P. Perret[11], A. Perro[47], K. Petridis[53], A. Petrolini[27,n], J. P. Pfaller[64], H. Pham[67], L. Pica[33,s], M. Piccini[32], B. Pietrzyk[10], G. Pietrzyk[13], D. Pinci[34], F. Pisani[47], M. Pizzichemi[29,p,47], V. Placinta[41], M. Plo Casasus[45], F. Polci[15,47], M. Poli Lener[26], A. Poluektov[12], N. Polukhina[42], I. Polyakov[47], E. Polycarpo[3], S. Ponce[47], D. Popov[7], S. Poslavskii[42], K. Prasanth[57], C. Prouve[45], V. Pugatch[51], G. Punzi[33,t], S. Qasim[49], Q. Q. Qian[6], W. Qian[7], N. Qin[4], S. Qu[4], R. Quagliani[47], R.I. Rabadan Trejo[55], J.H. Rademacker[53], M. Rama[33], M. Ramírez García[80], V. Ramos De Oliveira[68], M. Ramos Pernas[55], M.S. Rangel[3], F. Ratnikov[42], G. Raven[37], M. Rebollo De Miguel[46], F. Redi[28,i], J. Reich[53], F. Reiss[61], Z. Ren[7], P.K. Resmi[62], R. Ribatti[48], G. R. Ricart[14,81], D. Riccardi[33,s], S. Ricciardi[56], K. Richardson[63], M. Richardson-Slipper[57], K. Rinnert[59], P. Robbe[13], G. Robertson[58], E. Rodrigues[59], E. Rodriguez Fernandez[45], J.A. Rodriguez Lopez[73], E. Rodriguez Rodriguez[45], A. Rogovskiy[56], D.L. Rolf[47], P. Roloff[47], V. Romanovskiy[42], M. Romero Lamas[45], A. Romero Vidal[45], G. Romolini[24], F. Ronchetti[48], T. Rong[6], M. Rotondo[26], S. R. Roy[20], M.S. Rudolph[67], M. Ruiz Diaz[20], R.A. Ruiz Fernandez[45], J. Ruiz Vidal[79,aa], A. Ryzhikov[42], J. Ryzka[38], J. J. Saavedra-Arias[9], J.J. Saborido Silva[45], R. Sadek[14], N. Sagidova[42], D. Sahoo[74], N. Sahoo[52], B. Saitta[30,k], M. Salomoni[29,p,47], C. Sanchez Gras[36], I. Sanderswood[46], R. Santacesaria[34], C. Santamarina Rios[45], M. Santimaria[26,47], L. Santoro[2], E. Santovetti[35], A. Saputi[24,47], D. Saranin[42], A. Sarnatskiy[75], G. Sarpis[57], M. Sarpis[61], C. Satriano[34,u], A. Satta[35], M. Saur[6], D. Savrina[42], H. Sazak[16], F. Sborzacchi[47,26], L.G. Scantlebury Smead[62], A. Scarabotto[18], S. Schael[16], S. Scherl[59], M. Schiller[58], H. Schindler[47], M. Schmelling[19], B. Schmidt[47], S. Schmitt[16], H. Schmitz[17], O. Schneider[48], A. Schopper[47], M. Schubiger[36], N. Schulte[18], S. Schulte[48], M.H. Schune[13], R. Schwemmer[47], G. Schwering[16], B. Sciascia[26], A. Sciuccati[47], S. Sellam[45], A. Semennikov[42], T. Senger[49], M. Senghi Soares[37], A. Sergi[27,n], N. Serra[49], L. Sestini[31], A. Seuthe[18], Y. Shang[6], D.M. Shangase[80], M. Shapkin[42], R. S. Sharma[67], I. Shchemerov[42], L. Shchutska[48], T. Shears[59], L. Shekhtman[42], Z. Shen[6], S. Sheng[5,7], V. Shevchenko[42], B. Shi[7], Q. Shi[7], Y. Shimizu[13], E. Shmanin[42], R. Shorkin[42], J.D. Shupperd[67], R. Silva Coutinho[67], G. Simi[31,q], S. Simone[22,h], N. Skidmore[55], T. Skwarnicki[67], M.W. Slater[52], J.C. Smallwood[62], E. Smith[63], K. Smith[66], M. Smith[60], A. Snoch[36], L. Soares Lavra[57], M.D. Sokoloff[64], F.J.P. Soler[58], A. Solomin[42,53], A. Solovev[42], I. Solovyev[42], R. Song[1], Y. Song[48], Y. Song[4], Y. S. Song[6], F.L. Souza De Almeida[67], B. Souza De Paula[3], E. Spadaro Norella[28,o], E. Spedicato[23], J.G. Speer[18], E. Spiridenkov[42], P. Spradlin[58], V. Sriskaran[47], F. Stagni[47], M. Stahl[47], S. Stahl[47], S. Stanislaus[62], E.N. Stein[47], O. Steinkamp[49], O. Stenyakin[42], H. Stevens[18], D. Strekalina[42], Y. Su[7], F. Suljik[62], J. Sun[30], L. Sun[72], Y. Sun[65], D. Sundfeld[2], W. Sutcliffe[49], P.N. Swallow[52], F. Swystun[54], A. Szabelski[40], T. Szumlak[38], Y. Tan[4], M.D. Tat[62], A. Terentev[42], F. Terzuoli[33,w,47], F. Teubert[47], E. Thomas[47], D.J.D. Thompson[52], H. Tilquin[60], V. Tisserand[11], S. T'Jampens[10], M. Tobin[5,47], L. Tomassetti[24,l], G. Tonani[28,o,47], X. Tong[6], D. Torres Machado[2], L. Toscano[18], D.Y. Tou[4], C. Trippl[43], G. Tuci[20], N. Tuning[36], L.H. Uecker[20], A. Ukleja[38], D.J. Unverzagt[20], E. Ursov[42], A. Usachov[37], A. Ustyuzhanin[42], U. Uwer[20], V. Vagnoni[23], G. Valenti[23], N. Valls Canudas[47], H. Van Hecke[66], E. van Herwijnen[60], C.B. Van Hulse[45,y], R. Van Laak[48], M. van Veghel[36], G. Vasquez[49], R. Vazquez Gomez[44], P. Vazquez Regueiro[45], C. Vázquez Sierra[45], S. Vecchi[24], J.J. Velthuis[53], M. Veltri[25,x], A. Venkateswaran[48], M. Vesterinen[55], M. Vieites Diaz[47], X. Vilasis-Cardona[43], E. Vilella Figueras[59], A. Villa[23], P. Vincent[15], F.C. Volle[52], D. vom Bruch[12], N. Voropaev[42], K. Vos[76], G. Vouters[10,47], C. Vrahas[57], J. Wagner[18], J. Walsh[33], E.J. Walton[1,55], G. Wan[6], C. Wang[20], G. Wang[8], J. Wang[6], J. Wang[5], J. Wang[4], J. Wang[72], M. Wang[28], N. W. Wang[7], R. Wang[53], X. Wang[8], X. Wang[70], X. W. Wang[60], Y. Wang[6], Z. Wang[13], Z. Wang[4],

Z. Wang[28], J.A. Ward[55,1], M. Waterlaat[47], N.K. Watson[52], D. Websdale[60], Y. Wei[6],
J. Wendel[78], B.D.C. Westhenry[53], D.J. White[61], M. Whitehead[58], E. Whiter[52],
A.R. Wiederhold[55], D. Wiedner[18], G. Wilkinson[62], M.K. Wilkinson[64], M. Williams[63],
M.R.J. Williams[57], R. Williams[54], F.F. Wilson[56], W. Wislicki[40], M. Witek[39], L. Witola[20],
C.P. Wong[66], G. Wormser[13], S.A. Wotton[54], H. Wu[67], J. Wu[8], Y. Wu[6], Z. Wu[7],
K. Wyllie[47], S. Xian[70], Z. Xiang[5], Y. Xie[8], A. Xu[33], J. Xu[7], L. Xu[4], L. Xu[4], M. Xu[55],
Z. Xu[11], Z. Xu[7], Z. Xu[5], D. Yang, K. Yang[60], S. Yang[7], X. Yang[6], Y. Yang[27,n], Z. Yang[6],
Z. Yang[65], V. Yeroshenko[13], H. Yeung[61], H. Yin[8], C. Y. Yu[6], J. Yu[69], X. Yuan[5],
E. Zaffaroni[48], M. Zavertyaev[19], M. Zdybal[39], C. Zeng[5,7], M. Zeng[4], C. Zhang[6],
D. Zhang[8], J. Zhang[7], L. Zhang[4], S. Zhang[69], S. Zhang[6], Y. Zhang[6], Y. Z. Zhang[4],
Y. Zhao[20], A. Zharkova[42], A. Zhelezov[20], S. Z. Zheng[6], X. Z. Zheng[4], Y. Zheng[7],
T. Zhou[6], X. Zhou[8], Y. Zhou[7], V. Zhovkovska[55], L. Z. Zhu[7], X. Zhu[4], X. Zhu[8],
V. Zhukov[16], J. Zhuo[46], Q. Zou[5,7], D. Zuliani[31,q], G. Zunica[48]

[1]School of Physics and Astronomy, Monash University, Melbourne, Australia
[2]Centro Brasileiro de Pesquisas Físicas (CBPF), Rio de Janeiro, Brazil
[3]Universidade Federal do Rio de Janeiro (UFRJ), Rio de Janeiro, Brazil
[4]Center for High Energy Physics, Tsinghua University, Beijing, China
[5]Institute Of High Energy Physics (IHEP), Beijing, China
[6]School of Physics State Key Laboratory of Nuclear Physics and Technology, Peking University, Beijing, China
[7]University of Chinese Academy of Sciences, Beijing, China
[8]Institute of Particle Physics, Central China Normal University, Wuhan, Hubei, China
[9]Consejo Nacional de Rectores (CONARE), San Jose, Costa Rica
[10]Université Savoie Mont Blanc, CNRS, IN2P3-LAPP, Annecy, France
[11]Université Clermont Auvergne, CNRS/IN2P3, LPC, Clermont-Ferrand, France
[12]Aix Marseille Univ, CNRS/IN2P3, CPPM, Marseille, France
[13]Université Paris-Saclay, CNRS/IN2P3, IJCLab, Orsay, France
[14]Laboratoire Leprince-Ringuet, CNRS/IN2P3, Ecole Polytechnique, Institut Polytechnique de Paris, Palaiseau, France
[15]LPNHE, Sorbonne Université, Paris Diderot Sorbonne Paris Cité, CNRS/IN2P3, Paris, France
[16]I. Physikalisches Institut, RWTH Aachen University, Aachen, Germany
[17]Universität Bonn - Helmholtz-Institut für Strahlen und Kernphysik, Bonn, Germany
[18]Fakultät Physik, Technische Universität Dortmund, Dortmund, Germany
[19]Max-Planck-Institut für Kernphysik (MPIK), Heidelberg, Germany
[20]Physikalisches Institut, Ruprecht-Karls-Universität Heidelberg, Heidelberg, Germany
[21]School of Physics, University College Dublin, Dublin, Ireland
[22]INFN Sezione di Bari, Bari, Italy
[23]INFN Sezione di Bologna, Bologna, Italy
[24]INFN Sezione di Ferrara, Ferrara, Italy
[25]INFN Sezione di Firenze, Firenze, Italy
[26]INFN Laboratori Nazionali di Frascati, Frascati, Italy
[27]INFN Sezione di Genova, Genova, Italy
[28]INFN Sezione di Milano, Milano, Italy
[29]INFN Sezione di Milano-Bicocca, Milano, Italy
[30]INFN Sezione di Cagliari, Monserrato, Italy
[31]INFN Sezione di Padova, Padova, Italy
[32]INFN Sezione di Perugia, Perugia, Italy
[33]INFN Sezione di Pisa, Pisa, Italy
[34]INFN Sezione di Roma La Sapienza, Roma, Italy
[35]INFN Sezione di Roma Tor Vergata, Roma, Italy
[36]Nikhef National Institute for Subatomic Physics, Amsterdam, Netherlands
[37]Nikhef National Institute for Subatomic Physics and VU University Amsterdam, Amsterdam, Netherlands
[38]AGH - University of Krakow, Faculty of Physics and Applied Computer Science, Kraków, Poland
[39]Henryk Niewodniczanski Institute of Nuclear Physics Polish Academy of Sciences, Kraków, Poland
[40]National Center for Nuclear Research (NCBJ), Warsaw, Poland
[41]Horia Hulubei National Institute of Physics and Nuclear Engineering, Bucharest-Magurele, Romania
[42]Affiliated with an institute covered by a cooperation agreement with CERN
[43]DS4DS, La Salle, Universitat Ramon Llull, Barcelona, Spain
[44]ICCUB, Universitat de Barcelona, Barcelona, Spain
[45]Instituto Galego de Física de Altas Enerxías (IGFAE), Universidade de Santiago de Compostela, Santiago de Compostela, Spain



[46]Instituto de Fisica Corpuscular, Centro Mixto Universidad de Valencia - CSIC, Valencia, Spain

[47]European Organization for Nuclear Research (CERN), Geneva, Switzerland

[48]Institute of Physics, Ecole Polytechnique Fédérale de Lausanne (EPFL), Lausanne, Switzerland

[49]Physik-Institut, Universität Zürich, Zürich, Switzerland

[50]NSC Kharkiv Institute of Physics and Technology (NSC KIPT), Kharkiv, Ukraine

[51]Institute for Nuclear Research of the National Academy of Sciences (KINR), Kyiv, Ukraine

[52]School of Physics and Astronomy, University of Birmingham, Birmingham, United Kingdom

[53]H.H. Wills Physics Laboratory, University of Bristol, Bristol, United Kingdom

[54]Cavendish Laboratory, University of Cambridge, Cambridge, United Kingdom

[55]Department of Physics, University of Warwick, Coventry, United Kingdom

[56]STFC Rutherford Appleton Laboratory, Didcot, United Kingdom

[57]School of Physics and Astronomy, University of Edinburgh, Edinburgh, United Kingdom

[58]School of Physics and Astronomy, University of Glasgow, Glasgow, United Kingdom

[59]Oliver Lodge Laboratory, University of Liverpool, Liverpool, United Kingdom

[60]Imperial College London, London, United Kingdom

[61]Department of Physics and Astronomy, University of Manchester, Manchester, United Kingdom

[62]Department of Physics, University of Oxford, Oxford, United Kingdom

[63]Massachusetts Institute of Technology, Cambridge, MA, United States

[64]University of Cincinnati, Cincinnati, OH, United States

[65]University of Maryland, College Park, MD, United States

[66]Los Alamos National Laboratory (LANL), Los Alamos, NM, United States

[67]Syracuse University, Syracuse, NY, United States

[68]Pontifícia Universidade Católica do Rio de Janeiro (PUC-Rio), Rio de Janeiro, Brazil, associated to [3]

[69]School of Physics and Electronics, Hunan University, Changsha City, China, associated to [8]

[70]Guangdong Provincial Key Laboratory of Nuclear Science, Guangdong-Hong Kong Joint Laboratory of Quantum Matter, Institute of Quantum Matter, South China Normal University, Guangzhou, China, associated to [4]

[71]Lanzhou University, Lanzhou, China, associated to [5]

[72]School of Physics and Technology, Wuhan University, Wuhan, China, associated to [4]

[73]Departamento de Fisica, Universidad Nacional de Colombia, Bogota, Colombia, associated to [15]

[74]Eotvos Lorand University, Budapest, Hungary, associated to [47]

[75]Van Swinderen Institute, University of Groningen, Groningen, Netherlands, associated to [36]

[76]Universiteit Maastricht, Maastricht, Netherlands, associated to [36]

[77]Tadeusz Kosciuszko Cracow University of Technology, Cracow, Poland, associated to [39]

[78]Universidade da Coruña, A Coruna, Spain, associated to [43]

[79]Department of Physics and Astronomy, Uppsala University, Uppsala, Sweden, associated to [58]

[80]University of Michigan, Ann Arbor, MI, United States, associated to [67]

[81]Département de Physique Nucléaire (DPhN), Gif-Sur-Yvette, France

[a]Universidade de Brasília, Brasília, Brazil

[b]Centro Federal de Educacão Tecnológica Celso Suckow da Fonseca, Rio De Janeiro, Brazil

[c]Hangzhou Institute for Advanced Study, UCAS, Hangzhou, China

[d]School of Physics and Electronics, Henan University, Kaifeng, China

[e]LIP6, Sorbonne Université, Paris, France

[f]Excellence Cluster ORIGINS, Munich, Germany

[g]Universidad Nacional Autónoma de Honduras, Tegucigalpa, Honduras

[h]Università di Bari, Bari, Italy

[i]Università di Bergamo, Bergamo, Italy

[j]Università di Bologna, Bologna, Italy

[k]Università di Cagliari, Cagliari, Italy

[l]Università di Ferrara, Ferrara, Italy

[m]Università di Firenze, Firenze, Italy

[n]Università di Genova, Genova, Italy

[o]Università degli Studi di Milano, Milano, Italy

[p]Università degli Studi di Milano-Bicocca, Milano, Italy

[q]Università di Padova, Padova, Italy

[r]Università di Perugia, Perugia, Italy

[s]Scuola Normale Superiore, Pisa, Italy

[t]Università di Pisa, Pisa, Italy

[u]Università della Basilicata, Potenza, Italy

[v]Università di Roma Tor Vergata, Roma, Italy

[w]Università di Siena, Siena, Italy

$^{x}$*Università di Urbino, Urbino, Italy*
$^{y}$*Universidad de Alcalá, Alcalá de Henares, Spain*
$^{z}$*Facultad de Ciencias Fisicas, Madrid, Spain*
$^{aa}$*Department of Physics/Division of Particle Physics, Lund, Sweden*

$^{†}$*Deceased*