# Peer review of "Measurement of exclusive $J/ψ$ and $ψ(2S)$ production at $\sqrt{s}=13$ TeV"

_SciPost Physics, doi:SciPost Phys. 18, 071 (2025)_

## Round 1 · Referee Report · Anonymous (Referee 1) · 2024-10-24

Strengths
1- Improvement over previous results. 2- Results derived for the first time. 3- Detailed explanations and numerical results. 4- Useful comparisons with the theory and other results. 5- Well structured paper.
Weaknesses
1- Good fraction of the strategy very similar to that of the same analysis on a smaller data set at the same energy. 2- The highlight of the improvements this paper brings could be made clearer throughout the paper. 3- The presentation of previous results could be improved. 4- More references on the reconstruction and selection methods could be added.
Report
Even though it closely follows the strategy of a previous analysis on the same measurements at the same energy but with a smaller data set, it manages to present refinements over the previous results. More specifically, the cross section measurement for the central exclusive production of $J/\psi \rightarrow \mu^+\mu^-$ and $\psi(2S) \rightarrow \mu^+\mu^-$ is more precise, due to the utilization of a larger data set and, most importantly, to improvements in the method, leading to smaller systematic uncertainties. The larger data set also permits the measurement of the differential cross section of $\psi(2S)$ with more precision and in more rapidity bins, allowing for a better test of the theoretical calculations, which seem to be significantly higher than the measurements in the paper. The paper also presents a new result, the dependence of the $J/\psi$ and $\psi(2S)$ cross-sections on total transverse momentum transfer, measured for the first time in $pp$ collision and comparing it with $ep$ results. Given these nice advances, the significance of the paper could be highlighted even clearer in the paper, starting from the abstract.
The detailed explanations and numerical results in the main text and in the appendices are highly appreciated. However, the presentation of the previous results as well as the reconstruction and selection techniques, could be improved with more references.
All in all, I recommend the paper for publication.
Requested changes
Some suggested changes to improve clarity of the paper:
- The abstract could be slightly modified to highlight why the paper results are important (wrt. Ref 35), mentioning that the results are more precise, owing to the larger data set and the improved methods.
- It may be worth it to review the presentation of previous analyses and results. Some examples:
- Ref. 34, which is an analysis on PbPb collisions, is mentioned in the sentence of pp collisions.
- The paragraph about the usage of photoproduction cross-sections to update PDF fits, make predictions, etc. could be generalized and made more objective to mention that it is not only LHCb results used in those studies.
- The clarity of the text would profit from referencing papers/notes/etc. of the reconstruction and selection methods used by LHCb. Some examples:
- What is a muon of "good quality"?
- What is the HeRSCheL response (maybe this could be solved by re-citing Ref. [35]).
- To better understand the quality of the fits, the plots of fits overlaid on data points could include a panel on the bottom displaying the pull of the fit wrt. data.
- The presentation of Fig. 6 can be confusing, so it may make sense to slightly rework it. Some examples:
- Fig.6, right: How can it be claimed that the SPD multiplicity distributions "match reasonably well", where there seems to be a non-negligible mismatch in the shape. Maybe this mismatch is not important for the results of the analysis but this needs to be explained. Some quantification of "reasonably well" and why it is acceptable would be helpful.
- Fig. 6, left: The difference between the data with and without the HeRSCheL vetoes seems large for a good fraction of the plot. In the text, you claim that only the low $p_T^2$ range (how is that range defined?) is relevant. Then what is the purpose of showing the full range of the plot? It can distract the reader from the range that is important.
Recommendation
Publish (meets expectations and criteria for this Journal)
Author: Patrick Koppenburg on 2025-01-08 [id 5095]
(in reply to Report 1 on 2024-10-24)
Requested changes: - The abstract could be slightly modified to highlight why the paper results are important (wrt. Ref 35), mentioning that the results are more precise, owing to the larger data set and the improved methods.
REPLY: We usually do not refer to previous publications in abstracts, but indeed emphasised the changes in the introduction.
- It may be worth it to review the presentation of previous analyses and results. Some examples:
- Ref. 34, which is an analysis on PbPb collisions, is mentioned in the sentence of pp collisions.
REPLY: Good spot. Thanks for noticing that. The reference has been moved to the references of PbPb collisions.
- The paragraph about the usage of photoproduction cross-sections to update PDF fits, make predictions, etc. could be generalized and made more objective to mention that it is not only LHCb results used in those studies.
REPLY: This is not the place to review the inputs to PDF fits. What is relevant in this context are the measurements in the forward direction. The sentences has been rephrased accordingly.
- The clarity of the text would profit from referencing papers/notes/etc. of the reconstruction and selection methods used by LHCb. Some examples:
- What is a muon of "good quality"?
REPLY: We added citations in sections 2 and 4.
- What is the HeRSCheL response (maybe this could be solved by re-citing Ref. [35]).
REPLY: We now refer to https://arxiv.org/abs/1801.04281 where this response is described in detail.
- To better understand the quality of the fits, the plots of fits overlaid on data points could include a panel on the bottom displaying the pull of the fit wrt. data.
REPLY: We do have such plots for internal use, but do not include them in publications, as part of our LHCb style. In particular here we show 1D projections of 2 2D fit, so the relevant quantity to look at is the pull in 2D space. Such plots are provided in the supplementary material, in Fig.7, at https://cds.cern.ch/record/2909312/files/
- Fig.6, left: How can it be claimed that the SPD multiplicity distributions "match reasonably well", where there seems to be a non-negligible mismatch in the shape. Maybe this mismatch is not important for the results of the analysis but this needs to be explained. Some quantification of "reasonably well" and why it is acceptable would be helpful.
REPLY: Indeed, this is what the alternative modellings done in systematic checks aim at assessing. We clarify and forward-reference to Sec.5.
- Fig. 6, right: The difference between the data with and without the HeRSCheL vetoes seems large for a good fraction of the plot. In the text, you claim that only the low p2T range (how is that range defined?) is relevant. Then what is the purpose of showing the full range of the plot? It can distract the reader from the range that is important.
REPLY: We clarified that. The efficiency is determined at pt^2~0, which we clarify in the text. We also zoom into the region pt^2<0.4 GeV^2 in the plot.
Author: Patrick Koppenburg on 2025-01-08 [id 5096]
(in reply to Report 3 on 2024-11-09)How do we know that the inelastic p2T distribution is well-described below 0.9 GeV2? This is one of the leading sources of systematic uncertainty, and it is unclear how we know that the power-law model holds well in the signal region. The authors reference the H1 experiment measurement (arXiv:1304.5162), but in the H1 paper, the data in Fig. 6b shows some deviation from the model. It might help to expand the control region, defined by inverting the HeRSCheL veto, below 0.9 GeV2.
REPLY: We thank the referee for the suggestion. The data below 0.9 GeV2 and with inverted Herschel veto are not very instructive. They are dominated by CEP signal that escapes the veto due to noise. Because of this, there is no statistically significant information to gain about PD data. Using this region in the PD fit would require modelling the CEP component, which would increase the correlation of PD and CEP fit parameters. It is thus preferable to have a signal-free control sample. We added a phrase stating this.
Requested changes:
Please clarify the potential impact of mis-modeling the inelastic background at low p2T may have on the measurement.
REPLY: Indeed, we use the H1 model as an inspiration and found it provides the best fit. However in Sec. 5 we try alternative models, one of which (the sum of two exponentials) works well enough to provide an alternative solution. We expanded the text in Sec.5 to explain the rationale. The difference between the two fits is taken as systematic uncertainty.

---

## Round 1 · Referee Report · Anonymous (Referee 2) · 2024-10-25

Report
Requested changes
I do have a few questions or comments on some points, and resolving these prior to publication may improve the clarity of the paper:
-
near the bottom of page 4, a two-dimensional fit is introduced. It seems likely that the two dimensions are the dimuon mass and pt^2 mentioned in the previous sentence, but it would be good to state this explicitly.
-
l 9 on p 5: “their result are used” -> “their results are used”
-
in the 3rd paragraph on p5, it is stated that the difference in means between the Gaussians used to model the J/psi and psi(2s) signals is fixed the the known difference in the masses. Presumably the two peaks are not constrained individually to allow for the possibility of miscalibration of the masses. But in that case, wouldn’t it make more sense to constraint the ratio of the masses rather than the absolute difference?
-
on p6 it is stated that the fitted J/psi\gamma masses are shifted buy 6-10 MeV from their known values. It would be good to include the statistical uncertainty on these shifts.
-
for Figure 5 (and the similar figures in the appendix): is there a goodness-of-fit measure?
-
on p 8, it would be good to give a typical value (or range of values) for the SPD trigger inefficiency
-
Tables 1 and 2: it’s not clear why some value are blank. Are these the ones that are smaller than 0.005%? If so, explicity listing “<0.005%” would be helpful.
-
on p11, the1.5 sigma compatibility is wrt Ref 35? Some rewording could make this clearer.
-
One of the more striking outcomes of this measurement is that the differential cross section for J/psi agrees with an NLO calculation, but the psi(2S) differential cross section does not. Some comment on the potential implications of this would be welcome.
-
In the conclusion, when stating “their rapidity-dependent ratio” it would be helpful to clarify what “their” is referring to. “consistent but more precise than” -> “consistent with and more precise than”
Recommendation
Publish (easily meets expectations and criteria for this Journal; among top 50%)
Author: Patrick Koppenburg on 2025-01-08 [id 5097]
(in reply to Report 2 on 2024-10-25)
- near the bottom of page 4, a two-dimensional fit is introduced. It seems likely that the two dimensions are the dimuon mass and pt^2 mentioned in the previous sentence, but it would be good to state this explicitly.
REPLY: Thanks for the suggestion. It was added in page 4.
- l 9 on p 5: “their result are used” -> “their results are used”
REPLY: Done
- in the 3rd paragraph on p5, it is stated that the difference in means between the Gaussians used to model the J/psi and psi(2s) signals is fixed the the known difference in the masses. Presumably the two peaks are not constrained individually to allow for the possibility of miscalibration of the masses. But in that case, wouldn’t it make more sense to constraint the ratio of the masses rather than the absolute difference?
REPLY: The reason for the constraint is to avoid stochastic fluctuations in the psi(2S) mass due to low statistics. The J/psi mass is offset by -3 to +3 MeV from low to high rapidity, for the reasons you mention. The psi(2S) peak us thus expected to be offset by a similar amount. A proportional scaling would result in a shift of -3.5 to +3.5 MeV instead of -3 to +3, which has a negligible effect on the fit. The resulting differences are now reported in the table of systematic uncertainties.
- on p6 it is stated that the fitted J/psi\gamma masses are shifted by 6-10 MeV from their known values. It would be good to include the statistical uncertainty on these shifts.
REPLY: The statistical uncertainties are between 0.5 and 1 MeV depending on the bin. This information has been added.
- for Figure 5 (and the similar figures in the appendix): is there a goodness-of-fit measure?
REPLY: We do check the chi2/dof in the 2D pull as FigS7c.pdf in LHCb-PAPER-2024-012-supplementary.zip at https://cds.cern.ch/record/2909312/files/ . Depending on the y bin the chi2 varies between 0.9 and 1.3.
- on p 8, it would be good to give a typical value (or range of values) for the SPD trigger inefficiency
REPLY: We usually do not provide tables of efficiencies as they need long explanations of how the denominator is defined for each efficiency. In the case of the SPD efficiency it is 95%, independent of the rapidity range and year of data taking.
- Tables 1 and 2: it’s not clear why some value are blank. Are these the ones that are smaller than 0.005%? If so, explicitly listing “<0.005%” would be helpful.
REPLY: This is stated in the caption.
- on p11, the 1.5 sigma compatibility is wrt Ref 35? Some rewording could make this clearer.
REPLY: Stated that explicitly
- One of the more striking outcomes of this measurement is that the differential cross section for J/psi agrees with an NLO calculation, but the psi(2S) differential cross section does not. Some comment on the potential implications of this would be welcome.
REPLY: We have reworded this, emphasising the fact that the J/psi "prediction" is dated after the 2018 LHCb measurement, while the psi(2S) prediction from 2013 has not been revisited since.
- In the conclusion, when stating “their rapidity-dependent ratio” it would be helpful to clarify what “their” is referring to. “consistent but more precise than” -> “consistent with and more precise than”
REPLY. Thanks. 'Their' refers to the psi(2S)/J/psi ratio. This has been implemented.

---

## Round 1 · Referee Report · Anonymous (Referee 3) · 2024-11-9

Strengths
- The paper presents a new measurement of the central exclusive production of charmonium mesons, marking a significant improvement over previous measurements.
- It provides a rigorous analysis of background contributions and systematic effects to support the new level of precision.
Weaknesses
- How do we know that the inelastic $p^2_T$ distribution is well-described below 0.9 $GeV^2$? This is one of the leading sources of systematic uncertainty, and it is unclear how we know that the power-law model holds well in the signal region. The authors reference the H1 experiment measurement (arXiv:1304.5162), but in the H1 paper, the data in Fig. 6b shows some deviation from the model. It might help to expand the control region, defined by inverting the HeRSCheL veto, below 0.9 $GeV^2$.
Report
Requested changes
- Please clarify the potential impact of mis-modeling the inelastic background at low $p^2_T$ may have on the measurement.
Recommendation
Ask for minor revision

---

## Round 3 · Author Response

We thank the referees for the useful comments that help making the paper clearer.

---

## Round 3 · List of Changes

Introduction: - Clarified the rationale of the selection of references for PDFs. - Explained how the larger samples improves the background control with respect to the measurement with 2015 data Detector, etc: - Added references for track fit and muon ID. Two-dimensional signal fit: - Clarified the definition of fit variables - Specified the mass uncertainties in J/psi gamma fit Efficiencies: - Refer to Sec.5 for the SPD mismodelling - Changed Fig.6(right) to show the most relevant pt^2 region Systematics: - Specified how the psi(2S) is constrained. Following a suggestion from a reviewer a systematic uncertainty is added, which has a negligible effect. - Explained the choice of the pt^2 shape for the PD background Results: - Specified what 1.5sigma applied to - Expanded the comment on the psi(2S) prediction differing from the data. Conclusion: - Slight rewording as suggested by reviewers

---

## Editorial Decision

published